# Atmospheric oxidation drove climate change on Noachian Mars

Jiacheng Liu[1,2] ✉, Joseph R. Michalski [1] ✉, Zhicheng Wang[1] & Wen-Sheng Gao[3]

Modern Mars is bipolar, cold, and oxidizing, while early Mars was characterized by icy highlands, episodic warmth and reducing atmosphere. The timing and association of the climate and redox transitions remain inadequately understood. Here we examine the spatiotemporal distribution of the low surface iron abundance in the ancient Martian terrains, revealing that iron abundance decreases with elevation in the older Noachian terrains but with latitude in the younger Noachian terrains. These observations suggest: (a) low-temperature conditions contribute to surface iron depletion, likely facilitated by anoxic leaching through freeze-thaw cycles under a reducing atmosphere, and (b) temperature distribution mode shifted from elevation-dominant to latitude-dominant during the Noachian period. Additionally, we find iron leaching intensity decreases from the Early to Late Noachian epoch, suggesting a gradual atmospheric oxidation coupled with temperature mode transition during the Noachian period. We think atmospheric oxidation led to Mars becoming cold and bipolar in its early history.

The modern-day Martian climate is characterized by extreme cold and aridity, with temperature varying according to latitude and water ice primarily accumulating in the polar regions. Conversely, modeling results suggest that temperature was predominantly elevation-dominant and water ice mainly accumulated in highlands and polar regions regardless of the mean annual temperature (MAT) on early Mars[1–3]. Several potential mechanisms[4] have been proposed to drive the climate mode change, including the decrease in atmospheric $CO_2$[3,5] or atmospheric $H_2$[6,7], or a reduction in cloud radiative effect[6–8]. However, the exact process behind the climate mode change remains unclear. While significant cooling[7] and drying[9] have been suggested from Noachian Mars to Hesperian Mars, the timing of this climate mode transition remains poorly constrained[10].

Although the present surface and atmosphere of Mars are oxidizing, emerging evidence points to the existence of reducing gases (e.g., $H_2$) in a $CO_2$-dominant atmosphere on early Mars[11–13]. A chemically reducing atmosphere on early Mars is supported by multiple sources of $H_2$, including volcanoes[14–16], serpentinization[17,18], and impacts[19,20]. A reducing atmosphere can generate a strong greenhouse effect, which could have played a significant role in warming the early

Martian climate[14,21–27]. The intensity of greenhouse warming is closely tied to the concentration of reducing gases in the atmosphere, therefore, the oxidation of the Martian atmosphere could theoretically result in cooling of Martian climate[13].

The Mars Odyssey Gamma-ray spectroscopical (GRS) data reveal that surface iron (Fe) abundance in the Noachian terrains is relatively low compared to the Hesperian and the Amazonian terrains, as well as the global surface average Fe abundance[28] (Fig. 1a). As there is no correlation with other elements that are expected to vary with Fe during normal igneous processes (such as the correlation between Fe and Si), planetary crustal evolution cannot explain the secular variation in Fe abundance[28], especially the low Fe abundance in the Noachian terrains. Because GRS remote sensing technique is only sensitive to the uppermost ~30 cm of the surface, it is possible that the Fe depletion in the Noachian terrains could have been caused by surface aqueous processes[28,29]. Fe mobility can be affected by acidity, aqueous chemistry, redox state, and temperature during aqueous alteration[28–31]. Therefore, the redox and climate transitions could have been captured by the distribution of Fe abundance at the surface of Mars, especially when there was communication between the atmosphere,

[1]Department of Earth Sciences and Laboratory for Space Research, The University of Hong Kong, Hong Kong, China. [2]NWU-HKU Joint Center of Earth and Planetary Sciences, Department of Earth Sciences, The University of Hong Kong, Hong Kong, China. [3]School of Earth Resources, China University of Geosciences, 430074 Wuhan, China. ✉e-mail: jcliu01@hku.hk; jmichal@hku.hk

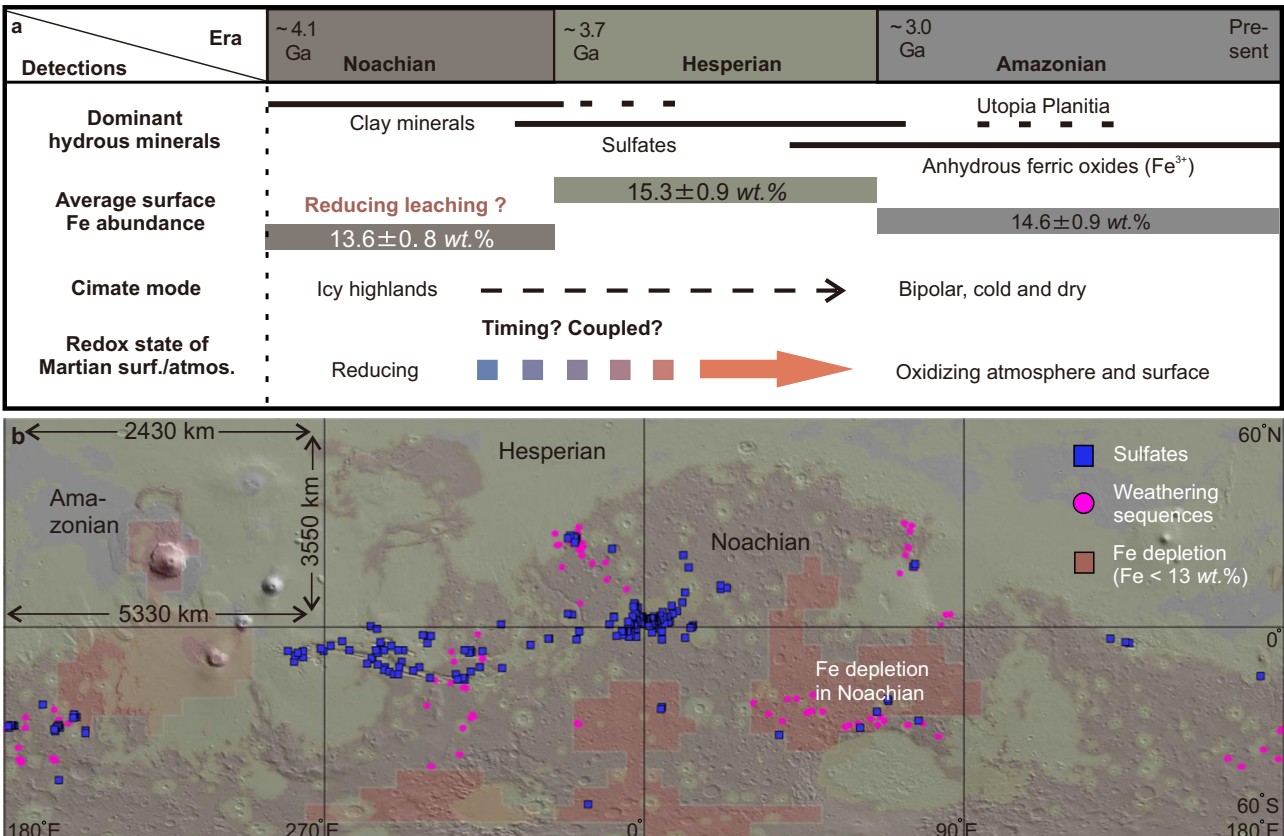

**Fig. 1 | The composition, climate, and redox transitions on Mars. a** Temporal transition of hydrous minerals, Fe abundance[28,55], climate and redox on Mars. **b** Spatial distribution of clays-rich paleosols[56] (pink cycles), sulfates[57] (blue squares), and Fe depletion region[13] (dark red area) on a Mars global map produced by Tanaka et al. (2014) [35] (https://pubs.usgs.gov/sim/3292/). The distance from 0 to 60°N/60°S is 3550 km. For each latitude of 90°, the distance is 5330 km in the equatorial region, but it decreases to 2430 km at 60°N/ 60°S.

hydrosphere, and lithosphere and efficient top-down leaching during the Noachian period. The characteristics of surface water can be recorded through Fe mobility; however, the Hesperian and Amazonian terrains do not exhibit signs of Fe depletion (Fig. 1a). Consequently, the study predominantly focuses on investigating the Noachian terrains.

Although low pH levels (pH < 3) can mobilize Fe, the acidic leaching scenario struggles to explain the Fe depletion across the extensive region of the surface (dark red area in Fig. 1b) from a mass balance perspective. This scenario is also inconsistent with the relatively high Th abundance found at the surface of the Noachian terrains[28], the solubility of which increases significantly with decreasing pH[32]. Furthermore, the absence of a positive correlation between Fe and Cl does not support the leaching of Fe as $FeCl_4^-$ complexes[33], which can transfer Fe efficiently. On the other hand, water redox can greatly influence Fe mobility. Specifically, Fe is immobile and tend to precipitate as Fe (oxyhydro)oxides under oxidizing conditions but becomes soluble and mobile as Fe(II) under reducing conditions. For example, recent studies have demonstrated that Fe loss through leaching as mobile Fe(II) was a common process in paleosols rich in clay minerals under a reducing atmosphere[11]. Consequently, Fe depletion is highly likely a result of anoxic chemical weathering and leaching under reducing conditions.

Anoxic chemical weathering and leaching can happen with the formation of clay minerals at temperatures well above freezing[11] or without the formation of clay minerals under temperatures near or slightly above 0 °C[34]. Although overlaps existed, there is no clear correlation between distributions of the Fe depletion zones (dark red area in Fig. 1b) identified through GRS data and paleosols rich in clay minerals (pink cycles in Fig. 1b) detected via infrared remote sensing using Compact Reconnaissance Imaging Spectrometer for Mars (CRISM) data. The decoupling is possibly related to the limited clay minerals' detections by the lack of areal coverage of CRISM data compared to the global coverage and larger footprint of GRS data. However, it is more likely that the Fe depletion was caused by reducing chemical weathering and leaching under temperatures around or just above 0 °C, which inhibits the formation of clay minerals[34].

This study aims to test whether the Fe depletion in the ancient Martian terrains resulted from icy weathering and leaching under reducing conditions by low-temperature processes, such as seasonal freeze-thaw cycles. A critical aspect of this research is the analysis of temporal and spatial distributions of Fe abundance to understand the timing of and the relationships among the redox and climate mode transitions. Our finding suggests that the surface temperature of Mars gradually evolved from an elevation-dominant to a latitude-dominant mode, coupled with atmospheric oxidation during the Noachian period.

## Results

### Spatiotemporal distribution of surface Fe abundance in the ancient Martian terrains

The global geologic map of Mars[35] (Supplementary Fig. 1) was used to identify the apparent relative surface age of each 5° × 5° bin of the GRS data[28]. In the Early and Middle Noachian terrains, the Fe abundance of regions with elevation above 3.0 km is approximately 14.0 *wt.%* (Fig. 2a and Supplementary Fig. 3a), which is close to global surface average (-14.3 *wt.%*). Between −3.0 to +3.0 km, Fe abundance shows a decreasing trend from -15.4 *wt.%* to -13.3 *wt.%* with increasing elevation (Fig. 2a). To be more specific, the decreasing trend is pronounced in

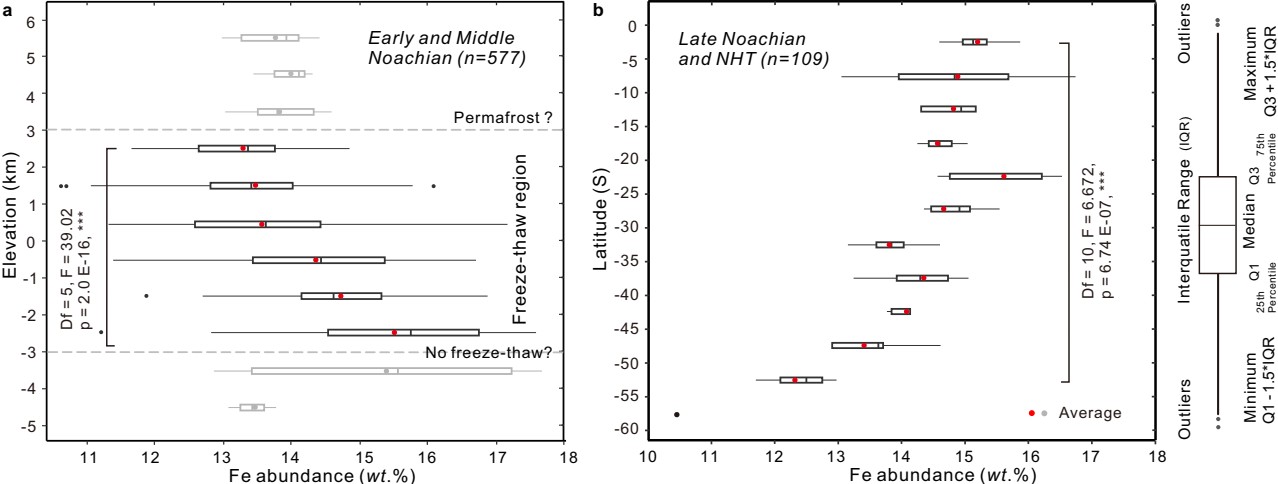

**Fig. 2 | The spatial evolution of Fe abundance at the surface of early Mars. a** The gradual decrease in Fe abundance with increasing elevation (−3.0 km – +3.0 km) at the surface of the Early and Middle Noachian terrains. **b** The gradual decrease in Fe abundance with latitude at the surface of the Late Noachian terrains and the Noachian-Hesperian transition (NHT) terrains in the southern hemisphere. Box plots indicate median (middle line), 25th, 75th percentile (box) and 95th percentile (whiskers) as well as outliers (single points). Source data are provided as a Source Data file.

tropics (20°S–20°N, Supplementary Figs. 4a, b and 5) and northern high latitudes (>20°N, Supplementary Fig. 4c, 5), both of which have elevations lower than 3 km. For southern high latitudes (>20°S), the average Fe abundance of regions with elevation above 3.0 km (-14.0 *wt.*%) is higher than that of regions between 0 and 3.0 km (-13.0 *wt.*%) (Supplementary Fig. 4d, 5). On the other hand, in the Late Noachian and Noachian-Hesperian transition (NHT) terrains, the surface Fe abundance is decorrelated with elevation (Supplementary Fig. 3a). It is worth noting that the correlation coefficient (r) between elevation and Fe abundance decreases gradually over time during the Noachian period (Fig. 3b).

Instead, there is a decreasing trend in Fe abundance with increasing latitude in the younger Noachian terrains (Fig. 2b and Supplementary Fig. 2b), especially on southern Mars with limited elevation variations. The correlation coefficient between Fe abundance and latitude is not strong in the Early Noachian terrains, but strong and positive in the Late Noachian (+0.52) and Noachian-Hesperian transition (+0.77) terrains of southern Mars (Supplementary Fig. 3b). It means that the correlation coefficient (r) increases gradually over time during the Noachian period (Fig. 3b).

The surface average Fe abundance in the Early, Middle, and Late Noachian terrains are all below the global surface average (Fig. 3a and Supplementary Fig. 6). Their values increase gradually with increasing amounts at -0.58 *wt.*% from the Early to the Middle Noachian terrains, -0.15 *wt.*% from the Middle to the Late Noachian terrains, and -0.10 *wt.*% from the Late Noachian terrains to the global surface average, respectively (Fig. 3b). However, the Fe abundance at the surface of the NHT terrains is remarkably high (-15.4 *wt.*%), which is -1.2 *wt.*% higher than that of the Late Noachian terrains and -1.1 *wt.*% higher than the global surface average (Fig. 3a).

**Statistical analyses**
ANOVA analyses indicate a significant disparity in Fe abundance across elevations ($F(5, 539) = 39.02$, $p < 0.001$) in the Early and Middle Noachian terrains and latitudes ($F(10, 61) = 6.67$, $p < 0.001$) in the Late Noachian and NHT terrains (Supplementary Tables 1–5). The analyzing results for different latitude bands (>20°N, 20°S–20°N, <20°S) can be found in Supplementary Tables 2–4). Also, ANOVA revealed a significant difference in Fe abundance across different age categories ($F(4, 810) = 41.37$, $p < 0.001$; Supplementary Table 6). Furthermore, the Z-statistic tests indicated statistical significance (95%

CI) in different age categories, except for Early Hesperian vs. NHT and Late Noachian vs. Middle Noachian comparisons (Supplementary Table 7).

## Discussion
It is logical to assume that higher elevations or higher latitude experienced colder temperatures[1,3]. Therefore, the correlation between surface Fe abundance with elevation or latitude in the Noachian terrains suggests that surface temperature may have influenced the Fe abundance at the surface. Fe abundance decreases with increasing elevations/latitudes (Fig. 2), suggesting that low-temperature conditions facilitated Fe depletion during the Noachian period.

Efficient Fe leaching under cooler conditions might have been related to elevated olivine weathering at cryogenic temperature[36,37]. Freezing can increase the acidity of residual water, potentially resulting in a pH level below 0, which counterintuitively accelerates the kinetics of chemical weathering to alter minerals at subfreezing temperatures[37,38]. For example, the weathering rate measured at −40 °C is comparable to forsterite dissolution rates in a pH of 2.5 solution at 25 °C[37]. In cold seasons, icy weathering can release Fe and form Fe-bearing salts[39]. During warm seasons, ice-blocked pores can become interconnected, allowing Fe-bearing salts to be dissolved by meltwater (Fig. 4a, b). Under reducing conditions, Fe is dissoluble and mobile as Fe(II). Due to the infiltration of meltwater containing Fe(II) (Fig. 4b, c), seasonal freeze-thaw can cause Fe leaching and therefore surface Fe depletion.

Under lower MAT, freezing can be amplified therefore can facilitate the release of Fe and subsequent leaching. This means that the Fe depletion is negatively related to temperature during seasonal freeze-thaw cycles. Therefore, the decreasing trend in Fe abundance with increasing elevation between −3.0 to + 3.0 km (Fig. 2a) may be a result of seasonal freeze-thaw cycles under reducing conditions. In addition, the top-down increasing trend of Fe abundance (Fig. 2a) can be partly contributed by the precipitation of leached Fe accumulated in low-elevation regions owing to intermittent oxidizing atmosphere during the Noachian period[13]. On the other hand, under temperatures well above freezing, chemical weathering can lead to direct top-down leaching (Fig. 4c). The rate of chemical weathering and Fe leaching is positively related to temperature, therefore more Fe can be depleted with decreasing elevation below -3 km, as indicated in Fig. 2a. The

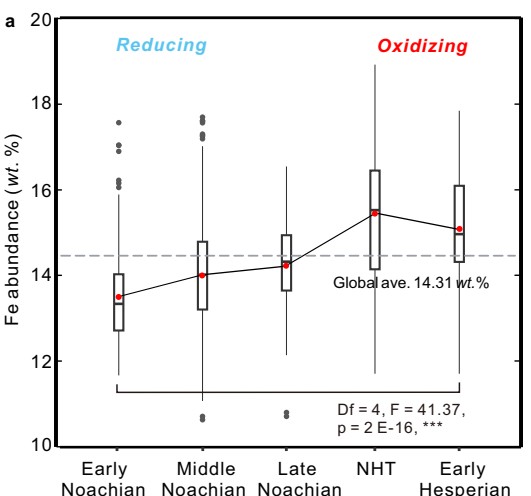

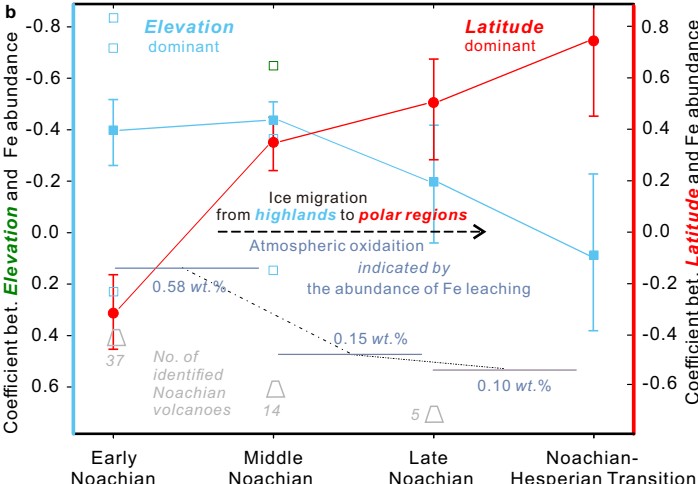

**Fig. 3 | The temporal evolution of Fe abundance at the surface and atmospheric redox on early Mars. a** The gradual increase in surface Fe abundance from the Early Noachian terrains to the Noachian-Hesperian transition (NHT) terrains. **b** The gradual decorrelation between Fe abundance and elevation (solid squares) but a gradual increasing correlation between Fe abundance and latitude (solid cycles) on Mars over time, suggesting a temperature distribution mode transition from elevation-dominant to latitude-dominant during the Noachian period. The climate transition is coupled with the gradual oxidation of the Martian atmosphere

indicated by the Fe leaching intensity. The abundance of Fe leaching in each epoch is calculated based on the value of Fe depletion relative to adjacent unit (for Early Noachian and Middle Noachian epoch) or the potential original Fe abundance of the Noachian crust (for Late Noachian epoch). Number of Noachian volcanoes was identified by Xiao et al.[18]. The hollow squares represent the correlation coefficient in three latitudinal bands (>20°N, 20°S-20°N, and < 20°S) of the Early and Middle Noachian terrains. More details can be found in Supplementary Fig. 5. Source data are provided as a Source Data file.

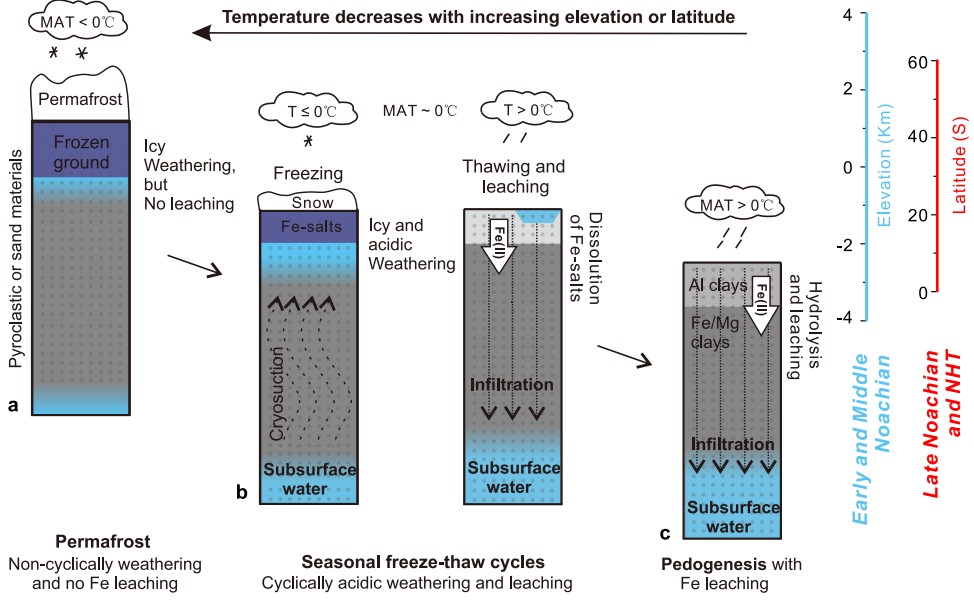

**Fig. 4 | Chemical weathering and Fe leaching happened at different temperatures during the Noachian period. a** In the high elevation/latitude region with mean annual temperature (MAT) below 0 °C, permafrost has existed with no complete thawing to leach Fe downwards. **b** In the middle elevation/latitude region with MAT around 0 °C, icy weathering during freezing can release significant Fe,

and the complete thawing during warm seasons can lead to significant leaching of the released Fe. Icy weathering under lower temperature can result in more Fe(II) release therefore more Fe leaching. **c** In the low elevation/latitude region with MAT above 0 °C, there are direct chemical weathering and Fe leaching the rates of which are positively correlated with temperature.

transition in chemical weathering rate and leaching efficiency partly accounts for the broad distribution range of Fe abundance at regions with elevation between −3 km and −4 km (Fig. 2a).

In the presence of permafrost, when rock pores are fully occupied by water ice, surface water is unable to directly connect with the subsurface, therefore dissolved Fe cannot be leached downwards (Fig. 4a). On Earth, permafrost typically forms in regions with a MAT below 0 °C[40]. Although an increasing leaching trend is observed with increasing elevation from -3 km to 3 km, the Fe abundance is relatively stable at ~14 *wt.%* in regions with elevations above 3 km

(Fig. 2a). This Fe abundance is close to the global surface average, suggesting limited top-down Fe leaching above 3 km. As the seasonal freeze-thaw region has efficient Fe leaching but the permafrost region has minimal Fe leaching, the transition in the Fe abundance trend at -3.0 km (Fig. 2a) suggests a shift from seasonal freeze-thaw to permafrost conditions. This suggests that the highlands with elevations above 3.0 km likely experienced permafrost during the Early and Middle Noachian epochs. Most bins with elevation above 3.0 km are in the southern high-latitude region (Supplementary Fig. 5). The minimal Fe leaching in permafrost can also explain the lack of

correlation between Fe depletion and elevation in southern high latitudes (Supplementary Fig. 4).

Permafrost was limited to regions with elevation above 3 km, suggesting a relatively restricted distribution of permanent ice deposits. This may explain why glacial features have not been extensively identified in ancient terrains[3]. Above 3.0 km, subglacial erosion[41] could form channels, while proglacial fluvial channels could develop below 3.0 km[42] during the Noachian period. The high elevation of the permafrost suggests that Mars has a warm climate with a global MAT > 0 °C during the Early and Middle Noachian epochs.

The statistical results reveal a wide range of Fe abundance within the same latitude, elevation, and epoch (Figs. 2, 3). These large variations may stem from several factors: (1) The difference in protolith, including differences in composition and resistance to chemical weathering; (2) Variations in temperature, therefore the Fe leaching intensity owing to different topography, and difference in latitude/elevation in the same elevation/latitude; (3) The sand/dust cover with a Fe abundance average of ~14.9 $wt.\%$[43], which is higher than the surface Fe abundance of Noachian terrains; (4) The large area of each GRS bin, which may introduce uncertainty in the Fe abundance of specific epoch, owing to several potential units with varying ages and Fe abundances in single bin.

The observation of less Fe at the surface of older Noachian Martian terrains (Fig. 3a) is consistent with irreversible and progressive Fe leaching under a reducing atmosphere[30]. Martian meteorite ALH84001, with an igneous age of $4.091 \pm 0.030$ billion years, is the only meteorite that can be used to constrain the composition of Martian Noachian crust. The Fe abundance of meteorite ALH84001 is ~14.3 $wt.\%$[44,45], which is similar to the global surface average Fe abundance based on GRS result. Assuming ~14.3 $wt.\%$ represents the original Fe abundance of the Noachian crust, the difference (~0.10 $wt.\%$) between this value and the Fe abundance of the Late Noachian terrains can represent the Fe leaching intensity during the Late Noachian epoch (Fig. 3b). However, for determining the abundance of Fe leaching specifically in the Early Noachian epoch, we cannot simply use the difference between Fe abundance of the Early Noachian terrains and the potential original Fe abundance of the Noachian crust, as this would represent the abundance of Fe leaching throughout the entire Noachian period rather than solely during the Early Noachian epoch. To determine the abundance of Fe leaching specifically in the Early Noachian epoch, we must subtract the Fe abundance of the Middle Noachian terrains from that of Early Noachian terrains. We employ the same method to calculate the abundance of Fe leaching in the Middle Noachian epoch. The resulted Fe leaching intensities in the Early and Middle Noachian epochs are approximately 0.58 $wt.\%$ and 0.15 $wt.\%$, respectively (Fig. 3b). They are minimum values of Fe depletion because the covering of sand and dust, which has higher average Fe abundance (~14.9 $wt.\%$)[43] than current Noachian surface average (~13.6 $wt.\%$, Fig. 1a), would cause the actual depletion values to be underestimated. Because older terrains might host more of these materials due to larger and more craters and longer time of sand/dust accumulation, the amounts of surface Fe depletion of the older Noachian epochs are possibly more underestimated. Therefore, excluding the sand/dust cover would strengthen the observed trend of decreasing Fe leaching intensity over time during the Noachian period (Fig. 3b).

Fe has been intensely leached during the Early Noachian epoch (Fig. 3b), suggesting its atmosphere was largely reducing. As volcanic outgassing is an important source of reducing gases[16], the most reducing condition in the Early Noachian epoch is consistent with the observation that more volcanoes observed in the Early Noachian terrains than younger Noachian terrains[43]. In addition, the thermal events linked to volcanic activities can promote groundwater circulation and result in widespread serpentinization[46], which can also produce abundant $H_2$[17]. Serpentinization might have produced significantly more reducing gases than previously thought[18] owing to the high Fe

abundance in the Martian crust[18]. Notably, the Fe abundance in the NHT terrains (Fig. 3b) is higher than 14.3 $wt.\%$, implying that Fe was enriched rather than leached during the NHT. This enrichment could have been the result of the efficient oxidation of the dissolved Fe(II) to deposit ferric minerals at the surface, suggesting a significant global atmosphere redox shift from reducing to oxidizing during the NHT. The redox shift may be related to a significant decrease in impact events, which is able to produce abundant $H_2$[19,20], after Late Heavy Bombardment.

These observations suggest a global and broad trend of oxidation, but they do not necessarily imply a linear transition from a reducing to an oxidizing environment. In fact, most of the reducing gases outgassing events (e.g., volcanoes and impacts) are intermittent, which more likely contribute to an episodic reducing atmosphere and episodic warm climate, as indicated by modeling result[13]. Additionally, the episodic reducing atmosphere and warm climate are more consistent with wet-dry cycling[47] and intermittent appearance of ferrous hydrous minerals at Gale crater[48,49], as identified by the Curiosity rover. This also aligns with short-term chemical weathering events occurring under warm conditions[50].

The observed negative correlation between Fe abundance and elevation during the Early and Middle Noachian epochs (Fig. 3b and Supplementary Fig. 3a) suggests that the temperature distribution was elevation dominant. The correlations are not very strong possibly due to the influence of bins with elevations above +3 km or below -3 km (Supplementary Fig. 3a) and the persistent effect of latitude on temperature and therefore Fe distribution during the Early and Middle Noachian epochs. Regardless of the relatively low r-value (Fig. 3b), the decreasing trend of r-value between Fe abundance and elevation over time makes sense. In contrast, the correlation between Fe abundance and latitude becomes stronger over time from the Early Noachian to Noachian-Hesperian transition (Fig. 3b). The patterns suggest that the Noachian period has experienced a gradual climate mode transition from an elevation-dominant temperature distribution (EDD) to a latitude-dominant distribution (LDD) (Fig. 3b).

Wordsworth's modeling work[1,3] suggests that temperature is correlated with elevation if there was high atmospheric pressure (e.g., ~1 bar surface pressure) and that temperature is decorrelated with elevation but correlated with latitude when atmospheric pressure is reduced to ~0.125 bar. It should be noted that the model only considers $CO_2$ and $H_2O$ as atmospheric constituents. In this context, the transition from EDD to LDD can be driven by a decline in atmospheric pressure[3], which could be a result of the loss of the Martian magnetic field during the Noachian period.

In addition to atmospheric pressure, recent results by Fan et al.[6] and Kite et al.[7] suggest that the greenhouse warming effect could play an important role in diabatic cooling. Under the strong greenhouse, the surface is mainly heated by the atmosphere, which is controlled by advection and convection over the lowlands, resulting in the formation of cold traps associated with topography. Conversely, under the weak greenhouse, the surface is primarily heated by insolation and thus surface temperature decorrelates with elevation. In this case, the observed gradual decrease in diabatic cooling could be attributed to the gradual weakening of the atmospheric greenhouse effect[6].

Figure 3b shows that, from the Early to Late Noachian epoch, Fe leaching intensity decreases, covarying with coefficients of Fe abundance with latitude and elevation (Fig. 3b). The reduced Fe depletion (Fig. 3a) over time during the Noachian period can be caused by a warmer global mean temperature (Fig. 4) or a less reducing environment. The increased latitude dependence can be caused by a reduced $H_2$ greenhouse effect or a reduced $CO_2$ greenhouse effect (a thinner atmosphere)[1,3], or a low obliquity. The decreased elevation dependence can be caused by a reduced $H_2$ greenhouse effect[7] or less $CO_2$[6]. The overlapping interpretation of these three aspects is the decrease in the amount of atmospheric $H_2$ overtime during the Noachian period. In

other words, it is highly possible that the Martian climate mode shifted from EDD to LDD during the Noachian period (Fig. 3b) was mainly driven by the atmospheric redox transition. In addition, the Fe abundance jumps from the Late Noachian to NHT (Fig. 3a) suggest a significant atmospheric redox shift from reducing to oxidizing, which could have considerably diminished greenhouse warming. This is consistent with a 10 °C-cooling driven by non-$CO_2$ radiative forcing during the Noachian-Hesperian transition, as suggested by a recent study on the distribution of water channels[7]. In conclusion, the spatial and temporal surface Fe distribution suggests that atmospheric oxidation has contributed to not only climate cooling but also ice migration from southern highlands to polar regions on Noachian Mars.

Owing to diabatic cooling in the Early and Middle Noachian epochs, precipitations and water ice would preferentially accumulate in highlands and the south circumpolar region[1,2,7]. As the climate transitioned from EDD to LDD in the Late Noachian epoch and NHT, the ice migrated from Martian highlands, including equatorial and low-latitude highlands, to polar regions. The ice migration led to increased absorption of solar energy radiation owing to the albedo decrease in equatorial and low-latitude regions, thereby moderating the rate of climate cooling during the Noachian-Hesperian transition. The transitional epoch facilitated a prolonged phase of periodic melting and fluvial/lacustrine activities, which would have produced sufficient meltwater to fill the observed lakes[10]. The climate transition scenario is consistent with the temporal distribution of valley networks that formed between the Middle Noachian epoch and the NHT[51], and their predominant spatial distribution in the low-latitude regions[1,52].

The seasonal freeze-thaw region has transitioned from high-elevation areas to high-latitude regions as the climate mode has shifted with the gradual loss of H and $H_2$ in the atmosphere. With further cooling of Mars after NHT, seasonal freeze-thaw regions could have migrated from high-latitude to low-latitude regions during the Hesperian period. Owing to the drying of Noachian Mars by sequestration of water in the crust[9], Hesperian Mars was arid with much less top-down leaching. Cryosuction during freezing can induce the accumulation of water and ions from groundwater in the near-surface environments[53]. During the thawing stage, strong evaporation under oxidizing conditions during the Hesperian period, could have precipitated salts and ferric oxides[54]. The enrichment of sulfates at Meridiani can be well explained by seasonal freeze-thaw cycles[54] according to the geochemical and textural observations from the Opportunity rover. The evidence includes upward enrichment of S but depletion of Cl, the upper bleaching and brecciated zone and lower darker and less brecciated zone with a sudden horizontal boundary in a meter-scale section, and the platy structure with abundant coarse planar voids but little fine voids. During the Hesperian period, the intersecting regions with high surface temperature (low latitude region) and a shallow water table promoted seasonal freeze-thaw with the upward migration of groundwater via cryosuction, thus providing favorable conditions for the formation of sulfates[54]. This mechanism may explain why sulfates (blue squares in Fig. 1b) tend to be enriched in low-latitude regions.

## Methods

### Spatiotemporal analyses of surface Fe abundance

The global geologic map of Mars produced by Tanaka et al. (2014)[35] was used to identify the apparent relative surface age of each 5° × 5° bin of the GRS data[28]. Cells were then given a specific age value based on the assigned geologic unit: Early Noachian, Middle Noachian, Late Noachian, or Noachian-Hesperian transition (Supplementary Fig. 1). Most 5° × 5° bins were not entirely age homogeneous. We assigned the most areally dominant apparent relative surface age for each bin. The dominant apparent surface age should correspond to at least half of the bin to avoid imprecise age assignment[28]. We excluded the cells without a specific geologic unit covering half of the bin. Although there

is age uncertainty, the areal binning of our apparent surface age categories is considered robust[28].

We used the average elevation of the gridded topographic data from NASA's Mars Global Surveyor (MGS) Mars Orbiter Laser Altimeter (MOLA) (Smith et al., 2001) to represent the elevation of each bin. The correlation coefficient ($r$) and corresponding uncertainty (95% CI) between latitude, elevation, and Fe abundance at the surface of terrains of different epochs is analyzed (Supplementary Figs. 3 and 5).

Three latitudinal bands (>20°N, 20°S–20°N, <20°S) were further divided to examine the correlation between Fe abundance and elevation of the Early and Middle Noachian terrains to reduce the latitude effect (Supplementary Figs. 4 and 5). Owing to a lack of sufficient data, we did not decompose the Late Noachian or NHT terrains. Concerning the relationship between Fe abundance and latitude, the analysis was primarily concentrated on the southern hemisphere (Fig. 2b and Supplementary Fig. 2b) in order to minimize the elevation effect because of the significant elevation variation in the northern hemisphere.

**Statistical analyses.** For the statistical approach, the aggregation of gamma-ray spectra yields mean elemental abundance and box plot for each category. The Fe abundances approximately adhere to normal distribution. Therefore, we employed ANOVA and z-statistic tests to compare differences in elemental abundance averages. Comparisons that reject the null hypothesis are reported as "statistically significant" with the respective confidence interval for that determination.

## Data availability

Data required to complete this work include (1) The Fe concentration at the surface of Mars from Mars Odyssey Gamma Ray Spectrometer and (2) The elevation of the surface of Mars from Mars Orbiter Laser Altimeter (MOLA). Both two planetary datasets are available within the JMARS software provided by Arizona State University (https://jmars.asu.edu) or through the Planetary Data System (https://pds-geosciences.wustl.edu/). The data generated in this study are provided in the Source Data file. Source data are provided with this paper.

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

## Acknowledgements

The authors are grateful to the Mars Odyssey Gamma Ray Spectrometer team whose diligent efforts have produced remarkable results. The work has benefited from discussion with James W. Head III and his recent talks. This work was supported by the National Natural Science Foundation of China Grant (42302264 and 42241129), Hong Kong Research Grants Council General Research Fund (17306623 and 17311022), and Hong Kong Research Grants Council Collaborative Research Fund (7004-21G) and Research Impact Fund (R5043-19).

## Author contributions

J.L. conceptualized the study. J.L. analyzed the data and wrote the paper. J.M. supervised the work and improved the manuscript. Z.W. collected the Fe abundance. W.-S.G. performed the statistical analysis. All authors read and approved the final manuscript.

## Competing interests

The authors declare no competing interests.
