## [Peer Review File · Nature Communications]

Atmospheric oxidation drove climate change on Noachian MarsReviewer #1 (Remarks to the Author):

The authors perform an analysis of the abundance of surface iron in ancient terrain on Mars. They find that in the oldest terrain, iron abundances are lowest at high elevations, while in younger terrain, abundance is more closely correlated with latitude. Interpreting these results through the lens of chemical weathering, they argue that they are most consistent with a scenario where early Mars was initially warmer with a more reducing atmosphere, and then cooled, oxidized and lost its atmospheric mass over time. They also suggest that the chemical transition was coupled to a climate transition whereby Mars's surface ice migrated from highland equatorial regions to polar regions.

This is an interesting study with important conclusions that is certainly worthy of publication in Nature Communications. I have a few comments for the authors to take into account, all of which are minor. After this, I'll be happy to recommend publication.

Minor comments:

For future reference, line numbers in the manuscript would have made it easier to write this review.

"the acidic leaching scenario is inconsistent with ..."

It might also be added that achieving acidic conditions on the scales necessary to explain Fe depletion across large regions of the surface is extremely challenging from a mass balance perspective.

"Efficient Fe leaching under cooler conditions might have been related to elevated olivine weathering at cryogenic temperature. Freezing can increase the acidity of residual water, which counterintuitively accelerates the kinetics of chemical weathering..."

This is an interesting idea. I think it'd be helpful here to specify the pH ranges we're talking about, as acidic leaching is argued against as the main mechanism earlier in the text. Can the authors quantify the magnitude of the effect they're talking about here? Seems like it matters for the interpretation of their data.

"Lower MAT can amplify the freezing, which can sustain chemical weathering and make the residual water more acidic (Figure 4a). As a potentially counterintuitive result, more Fe can be released under colder conditions"

This text is a bit repetitive with the sentence I just quoted above. I suggest editing to improve the flow.

". The relatively limited distribution of the permanent ice deposits may explain why glacial features are not extensively identified in the southern highlands"

It's worth citing and discussing some of the more recent literature on this topic here, e.g.:

Grau Galofre, A., Jellinek, A.M. and Osinski, G.R., 2020. Valley formation on early Mars by subglacial and fluvial erosion. *Nature Geoscience*, 13(10), pp.663-668.

Boatwright, B.D. and Head, J.W., 2021. A Noachian proglacial paleolake on Mars: fluvial activity and lake formation within a closed-source drainage basin crater and implications for early Mars climate. *The Planetary Science Journal*, 2(2), p.52.

"Assuming the global average surface Fe abundance represents the original Fe abundance in the Noachian terrains"

I didn't quite follow the logic of this statement. The Fe content of e.g. Hesperian plains should be representative of the starting Noachian value only if they are sourced from mantle material with the same Fe content. Do we have independent constraints on this from mantle evolution modeling or other sources?

"A recent modeling result by Fan et al., (2022) indicates that the greenhouse warming effect could play a more important role than atmospheric pressure in diabatic cooling"

It's worth noting that the Fan et al. preprint mischaracterizes past work on this topic, as neither Wordsworth (2016) or the earlier Forget et al. (2013) claimed that sensible heat fluxes are solely responsible for adiabatic cooling of surfaces under a thicker atmosphere (see e.g., Wordsworth 2016; p. 396).

Fig. 1: The authors show sulfates on the plot but don't discuss them much in the text. Are their results consistent with observations of sulfur minerals on Noachian and Hesperian terrain?

Fig. 4: I strongly suggest changing "reducing greenhouse warming" to "decreasing greenhouse warming" here to avoid confusion.

Reviewer #2 (Remarks to the Author):

Please check the attached comments.

Reviewer #3 (Remarks to the Author):

Please check the attached .docx file.

Reviewer #2 and co-reviewer #3 Attachment on the following page

The paper analyzes remote sensing data to explore surface Fe abundance on ancient Martian terrains from the early Noachian to the Noachian-Hesperian Transition epoch. The authors find a gradual decrease of Fe depletion, as well as a transition of spatial pattern from altitude-dominated to latitude-dominated over time. The authors conclude that the Fe transition can be explained by the change of iron leaching intensity under a reducing environment, which is ultimately connected to a transition of atmospheric redox on early Mars.

This paper provides new geochemical evidence relevant to the climate change on early Mars. It contributes to a converging view among the Mars community, together with other data papers and modeling papers. However, as a climate modeler, I do have several major concerns on data analysis and literature review. I believe the paper will be much more convincing to climate scientists once the comments are resolved. Since these comments are straightforward to apply, I recommend minor revisions to the editor.

Major comments:

1) *The citations to climate papers needs to be improved.* For example, the authors state that the transition from icy highlands to the modern mode is not fully understood at Line 29-31, with the refence to James Head's conference talk. Yet the transition and mechanism were discussed in depth in Kite et al. (2022) and Fan et al. (2023). The other significant example is the citation to Wordsworth (2017) at Line 38. Yet Wordsworth (2017) is neither the earliest, nor the most precise paper on the greenhouse effect caused by a reducing atmosphere. Examples like these are numerous throughout the Introduction and Discussion. My suggestions to address this issue are listed below, with the reference list attached in the end.

Line 29: Early modeling results should cite Wordsworth (2016).

Line 29-31: The transition has been discussed in Kite et al. (2022) and Fan et al. (2023). There is not yet a consensus on which mechanism drives the change of greenhouse effect. It could be the decrease of CO₂ (Wordsworth, 2016; Kite, 2019), or the decrease of H₂ (Fan et al., 2023), or cloud radiative effect (Urata & Toon, 2013).

Line 32-34: There are other papers that suggest the presence of reducing gases based on the modern record. Examples include, for atmospheric evolution, Yoshida & Kuramoto (2020) and Wordsworth et al. (2021), and for surface river records, Kite et al. (2022).

Line 34-36: Please add the refences that quantified the potential source of H₂:

Volcanos: Ramirez et al. (2014); Batalha et al. (2015)

Serpentinization: Liu et al. (2021a, 2021b), Tutolo & Tosca (2023)

Impacts: Haberle et al. (2019)

Line 35-37: The reference to Tutolo & Tosca (2023) should be merged to serpentinization papers at Line 34-36. If the authors do think this paper needs more exposure, I recommend switching the sentence to Discussion.

Line 38: Using Wordsworth (2017) as the only one citation is unbalanced. The hydrogen greenhouse effect hypothesis can be traced at least as far back to Sagan (1977). The modern discussions were initiated by Wordsworth & Pierrehumbert (2013) and Ramirez et al. (2014). The most up-to-date data are Turbet et al. (2020), Godin et al. (2020), Kamada et al. (2021, 2022).

Line 39-41: If the authors want to keep reference 7-9, Kamada et al. (2021, 2022) should be added for their contribution in quantifying the greenhouse effect in GCMs.

Line 109-110: Cite Wordsworth (2016)

Line 199-200: Cite Kite (2022)

Line 219: Cite Kite (2022)

2) *Data analysis for the altitude dependence should be improved.* The term “icy highlands” can be confusing because under the “icy highlands” regime (or equivalently the strong greenhouse regime), both high altitude and high latitude are cold traps of a planet. For both endmembers of the temperature pattern, a large equator-to-pole temperature gradient ($dT > 30$ K) on early Mars is expected (see Figure 1 in Fan et al., 2023 for the example). If the authors plot all datapoints on a temperature-altitude plot, the correlation coefficient can be much smaller than it should be for a fixed solar heating due to the equator-to-pole insolation gradient. This issue can be much worse for scenarios with a small obliquity (this may be why Wordsworth, 2016 only show the case with obliquity = 41.8 deg).

My suggestion is to decompose the altitude effect and latitude effect for each epoch. For example, the authors could use three latitudinal bands (tropics, northern high latitudes, southern high latitudes) like Fan et al. (2023), or even more latitudinal bands (from low to high latitudes) like Faulk et al. (2017). I expect the authors will get higher coefficients for the altitude dependence during the early Noachian and Middle Noachian epoch. Even if the coefficients are still low, it offers more information, specifically that the insolation at one latitudinal band was disrupted by orbital variations.

For the latitudinal dependence, the authors may choose not to decompose the NHT or even the late Noachian epoch due to the lack of sufficient data. It is OK because the temperature pattern for the modern regime is 1-D. But again, it is expected that for early epochs the latitudinal coefficients can be larger once the altitudinal dependence is excluded.

Decomposing will make the manuscript's figures more complicated (e.g., Fig. S1). To keep the figures in the main text simple, possible modifications include (1) choosing a typical latitudinal band for the early epoch in Figure 2 and (2) replacing one green square to multiple green squares in Figure 3b.

Other comments:

Title: Please revise as "... on Noachian Early Mars" or "on Early Mars between 4.1 Ga – 3.7 Ga", for precision.

Abstract: Overall the statements are too strong. At the current stage of this paper, the transition of altitude-dependence needs to be further verified (see major comment #2); the role of igneous processes in changing iron abundance needs further work; the climate & redox interpretation to the temperature pattern is complex (see the comments on Discussion). With those uncertainties, the story implied from the work is still a hypothesis, not a fact. Recommend revisions include (1) changing "imply" at Line 12 to "suggest"; (2) changing the last sentence as "The change of iron abundance suggest ..." or "We think ...".

Line 46-47: Please justify the focus on the Noachian terrains for the later part of the paper (i.e., please explain why the conclusions cannot be extrapolated to Hesperian and Amazonian).

Line 51-58: A number of unpreferred interpretations are explained in this paragraph, but somehow the preferred one is not. It is surprising that "Fe leaching under a reducing condition" is explained late in Discussion. Since it is the key concept of the paper, please give a basic introduction here (with one or two sentences).

Line 60-67: The reasoning here may be hard to follow for some readers. What does the discrepancy from CRISM data say? Does that mean another mechanism for iron depletion? Why in the end the authors go back to the same conclusion of chemical weathering?

Line 60-63 & Figure 1:

- (1) Recommend revising the label & legend in Figure 1b alongside the text. Which part of Figure 1b is referred to exactly? Is it comparing the iron depletion zone (dark red?) and paleosols (brown? pink?)? Why shows sulfate?
- (2) Please include a geological map with different epochs of Noachian in the Supplementary Material.

Line 66: How low does the temperature need to be? Specifying the temperature range required here will be very helpful to quantifying the atmospheric concentration / greenhouse effect for future climate modeling works.

Line 77: Please give a basic description of the research approach first. Some readers might not know where “bins” come from.

Line 77-79: This goal of this paragraph may be confusing to some readers. Is it necessary to include it in the main text (rather than in the Methods)?

Figure 2:

1. Very interesting results! Could the authors please add more interpretation why the snowline can explain the transition of trend? Fan et al. (2023) predicts either a temperature discontinuity for the weak greenhouse case, or a smooth trend following an adiabat for the strong greenhouse case. Yet the data seems not applicable to either case. Why should data above snowline should have the average value? If chemical weathering does not take place under this regime, then should it not record the largest iron abundance? Please can the authors elaborate and/or explain?
2. Please can the exclusion of the -3.5 line be justified? For the freeze-thaw cycle, alternatively you can say the cycle extends to -4~-3 km and the transition at bottom is just a lack of sufficient data. If the authors wish to keep their current boundary of no freeze-thaw, it is recommended that they give additional discussion on how clay formation explains the iron abundance at low altitudes.
3. Please add the timing for each panel (early & middle → when? late and NHT → when?)
4. How does the figure look like when the altitude and latitude effect are fully decomposed? (see major comment #2). For example, how does latitude relation work under the early time? How does the elevation relation work under the late time? The authors might want to put most of the results into supplementary information...
5. Please add $n = ?$ for the number of the data included.

Figure S1:

Recommend add timing (following Figure 2)

Recommend decomposing the latitude and altitude effect (following Figure 2)

Figure 3a:

1. If my reading is correct, the difference between Early Noachian average (red dot) and global average (horizontal dashed line), which is 0.58, is not about four times larger compared to the difference between Middle Noachian average and global average, which is 0.15. Is there something wrong with data visualization?
2. How reducing would the atmosphere be (e.g., partial pressure of H₂) for iron depletion? Please add a citation (or calculation).
3. The light blue numbers and horizontal lines in Figure 3a are confusing and distracting. Since there is already a clear presentation of them in Figure 3b, I would recommend just removing them in Figure 3a.

Figure 3b: Please add error bars for the correlation analysis (same in Figure S1).

Tables:

1. Decompose the latitude and altitude effect (following Figure 2)
2. A missing “e” for “Latitud” in Table S2.

Line 122: There is no direct demonstration of infiltration or Fe leaching in Figure 3b (referring to somewhere else?).

Line 123-124: The necessity of a reducing environment has not been explicitly stated yet (“mobile iron = reducing” is unclear for atmospheric scientists).

Fig. 4:

1. Based on the reasoning in this Figure, the text in Figure 2a should be revised from “snow line” to “permafrost”.
2. Following the statement from Line 136-140, iron abundance stays high under a clay-forming regional climate. Inconsistently, all the strange signals (wide data range for -4~3 km & a transition of trend for -4~-5 km) are attributed to clay mineral formation. It remains unclear to me how the two stories can be reconciled.

Line 182 – 186: The iron enrichment during the late stage is important. Maybe move to Results?

Line 200-202: I don’t think this is true. A transition from ADD to LDD would imply a net decrease of the greenhouse effect, which means a lower MAT.

Line 212-239: Even taking the three aspects of iron abundance evolution to be true, the interpretations to climate & redox can be more complex than what the authors have shown. The less iron depletion over time can be caused by a **warmer global mean temperature**, or a **less reducing environment**. The more latitudinal dependence can be caused by a **thinner atmosphere**, or a **lower obliquity**, or **less H2 greenhouse effect**, or **less CO2 greenhouse effect**. The less altitude dependence can be caused by **less H2 greenhouse effect or less CO2**. Here red means a warming climate, cyan means a slightly colder climate, and blue means a much colder climate. The overlap of the interpretations on the three aspects is the conclusion of this paper, but the complexities/uncertainties for this logical step needs to be demonstrated.

References:

Kite, E. S., Mischna, M. A., Fan, B., Morgan, A. M., Wilson, S. A., & Richardson, M. I. (2022). Changing spatial distribution of water flow charts major change in Mars’s greenhouse effect. *Science Advances*, 8(21), eabo5894.

Fan, B., Jansen, M. F., Mischna, M. A., & Kite, E. S. (2023). Why are Mountaintops Cold? The Transition of Surface Lapse Rate on Dry Planets. *arXiv preprint arXiv:2311.10151*.

Wordsworth, R., Kalugina, Y., Lokshtanov, S., Vigasin, A., Ehlmann, B., Head, J., ... & Wang, H. (2017). Transient reducing greenhouse warming on early Mars. *Geophysical Research Letters*, *44*(2), 665-671.

Wordsworth, R. D. (2016). The climate of early Mars. *Annual Review of Earth and Planetary Sciences*, *44*, 381-408.

Urata, R. A., & Toon, O. B. (2013). Simulations of the martian hydrologic cycle with a general circulation model: Implications for the ancient martian climate. *Icarus*, *226*(1), 229-250.

Kite, E. S., Steele, L. J., Mischna, M. A., & Richardson, M. I. (2021). Warm early Mars surface enabled by high-altitude water ice clouds. *Proceedings of the National Academy of Sciences*, *118*(18), e2101959118.

Yoshida, T., & Kuramoto, K. (2020). Sluggish hydrodynamic escape of early Martian atmosphere with reduced chemical compositions. *Icarus*, *345*, 113740.

Wordsworth, R., Knoll, A. H., Hurowitz, J., Baum, M., Ehlmann, B. L., Head, J. W., & Steakley, K. (2021). A coupled model of episodic warming, oxidation and geochemical transitions on early Mars. *Nature Geoscience*, *14*(3), 127-132.

Ramirez, R. M., Kopparapu, R., Zugger, M. E., Robinson, T. D., Freedman, R., & Kasting, J. F. (2014). Warming early Mars with CO₂ and H₂. *Nature Geoscience*, *7*(1), 59-63.

Batalha, N., Domagal-Goldman, S. D., Ramirez, R., & Kasting, J. F. (2015). Testing the early Mars H₂-CO₂ greenhouse hypothesis with a 1-D photochemical model. *Icarus*, *258*, 337-349.

Haberle, R. M., Zahnle, K., Barlow, N. G., & Steakley, K. E. (2019). Impact degassing of H₂ on early Mars and its effect on the climate system. *Geophysical Research Letters*, *46*(22), 13355-13362.

Liu, J., Michalski, J. R., & Zhou, M. F. (2021). Intense subaerial weathering of eolian sediments in Gale crater, Mars. *Science Advances*, *7*(32), eabh2687.

Liu, J., Michalski, J. R., Tan, W., He, H., Ye, B., & Xiao, L. (2021). Anoxic chemical weathering under a reducing greenhouse on early Mars. *Nature Astronomy*, *5*(5), 503-509.

Tutolo, B. M., & Tosca, N. J. (2023). Observational constraints on the process and products of Martian serpentinization. *Science Advances*, *9*(5), eadd8472.

SAGAN, C. (1977). Reducing greenhouses and the temperature history of Earth and Mars. *Nature*, 269(5625), 224-226.

Wordsworth, R., & Pierrehumbert, R. (2013). Hydrogen-nitrogen greenhouse warming in Earth's early atmosphere. *science*, 339(6115), 64-67.

Turbet, M., Boulet, C., & Karman, T. (2020). Measurements and semi-empirical calculations of CO₂+ CH₄ and CO₂+ H₂ collision-induced absorption across a wide range of wavelengths and temperatures. Application for the prediction of early Mars surface temperature. *Icarus*, 346, 113762.

Godin, P. J., Ramirez, R. M., Campbell, C. L., Wizenberg, T., Nguyen, T. G., Strong, K., & Moores, J. E. (2020). Collision-induced absorption of CH₄-CO₂ and H₂-CO₂ complexes and their effect on the ancient Martian atmosphere. *Journal of Geophysical Research: Planets*, 125(12), e2019JE006357.

Kamada, A., Kuroda, T., Kasaba, Y., Terada, N., & Nakagawa, H. (2021). Global climate and river transport simulations of early Mars around the Noachian and Hesperian boundary. *Icarus*, 368, 114618.

Kamada, A., Kuroda, T., Kodama, T., Kasaba, Y., & Terada, N. (2022). Evolution of ice sheets on early Mars with subglacial river systems. *Icarus*, 385, 115117.

Reviewer #4 (Remarks to the Author):

Dear Jiacheng Liu et al.,

This was an exciting manuscript to read and some big impacts for the planetary community. In an attachment, I have provided a major comment with some more modest suggestions to help strengthen the manuscript. Please reach out if you need any clarification and good luck with the process.

Sincerely,
Mike Thorpe

Reviewer #4 Attachment on the following page

Dear Jiacheng Liu et al.,

This was an exciting manuscript to read and some big impacts for the planetary community. Below I have provided a major comment with some more modest suggestions to help strengthen the manuscript. Please reach out if you need any clarification and good luck with the process.

Sincerely,

Mike Thorpe (michael.t.thorpe@nasa.gov)

Major Comment:

The one major comment I have on this manuscript is to at least address or link some of these results to rover findings. As it is currently written, the processes seems very linear from a reducing to an oxidizing environment. But in Gale crater, we have examples within the stratigraphy that reduced phases and oxidized phases are fluctuating in a way that might be challenging to describe the paleoenvironment as a linear transition. Even more interesting is that we have reduced carbonates in the most recent drill holes, presumably highest in the stratigraphy. I feel this manuscript should address in situ measurements to strengthen any implications of this work.

Modest Comments/Suggestions:

Can you discuss outliers a bit and how statistically meaningful these trends are. It's a r max of 0.47 for elevation and 0.77 for latitude but that's also with far fewer points.

In line 89 the authors write "association between Fe abundance and latitude is weak." These weak associations have r values not too far off from elevation. While I still think the trends and tightness of the grouping makes sense, it is something to note here.

Line 126: The authors state a "negative correlation between Fe and Mg according to the global GRS" but only cite a reference. If this is a need to support the argument, this correlation should at least be portrayed in the supplementary. I think the authors are trying to suggest the closure effect and that since no other elements are increasing while Fe decreases, they need to support it with Mg rising. Also, in the lines above the authors address Fe vs. Mg olivine but another route for this trend could be that both phases are liberated during incipient alteration and then their behavior differs with increasing alteration. Could be another mechanism to address.

Line 153: How warm?

Line 159: The authors mention "sand cover increases the amount of Fe" but do not have values to support that. I think this is right but perhaps the manuscript could benefit from including values. One suggestion would be to cite the composition of Fe in sand vs. dust. Maybe use some examples from the rover and global dust values?

Line 172: The authors state “sand cover does not affect the observed trend of decreasing Fe leaching intensity over time”, however, could this trend also be because older sands have more time to accumulate dust?

Figure 1: the sulfates are not really discussed all that much in text. I suggest adding a couple lines in the text to warrant it's importance or delete the sulfates in this map because my eyes are drawn there.

Figure 3b: migration to poles vs. polars? Or polar regions vs. polars?

Check all figure numbers and references to figures throughout.

Manuscript Number: NCOMMS-23-54917

Title: Atmospheric Oxidation Drove Climate Change on Noachian Mars

Dear Reviewers,

We are grateful for your thoughtful reviews, which have not only guided us in enhancing the manuscript but also recognized the potential impact of this work. In response to your valuable comments, we revised the manuscript thoroughly and conducted a self-check. Additionally, we prepared a detailed, point-by-point response to the reviewers' feedback. In our response document, the **green text** represents our responses, while the **red text** indicates the corresponding changes made in the revised manuscript. We believe that this updated version of the manuscript addresses all concerns raised from you.

You can find our replies to your comments in the following pages:

Reviewer #1: page 2-5

Reviewer #2&3: page 6-22

Reviewer #: page 23-26

Thank you very much for your reviewing efforts!

Sincerely,

Jiacheng Liu

REVIEWER COMMENTS

Reviewer #1 (Remarks to the Author):

The authors perform an analysis of the abundance of surface iron in ancient terrain on Mars. They find that in the oldest terrain, iron abundances are lowest at high elevations, while in younger terrain, abundance is more closely correlated with latitude. Interpreting these results through the lens of chemical weathering, they argue that they are most consistent with a scenario where early Mars was initially warmer with a more reducing atmosphere, and then cooled, oxidized and lost its atmospheric mass over time. They also suggest that the chemical transition was coupled to a climate transition whereby Mars's surface ice migrated from highland equatorial regions to polar regions.

This is an interesting study with important conclusions that is certainly worthy of publication in Nature Communications. I have a few comments for the authors to take into account, all of which are minor. After this, I'll be happy to recommend publication.

Reply: Thank you very much for your encouragement and useful comments. We modified the manuscript carefully. We appreciate your guidance in enhancing our work.

Minor comments:

For future reference, line numbers in the manuscript would have made it easier to write this review.

Reply: We apologize for any inconvenience caused by the absence of line numbers in the version of the manuscript you reviewed. It appears that other reviewers were able to see the line number of the original manuscript based on their review reports. We hope that the line number are clearly visible in the updated version for your reference.

“the acidic leaching scenario is inconsistent with ...”

It might also be added that achieving acidic conditions on the scales necessary to explain Fe depletion across large regions of the surface is extremely challenging from a mass balance perspective.

Reply: Thank you for raising this point. We incorporated the point in lines 57-59: “Although low pH levels (pH<3) can mobilize Fe, the acidic leaching scenario struggle to explain the Fe depletion across the extensive region of the surface (dark

red area in Fig. 1b) form a mass balance perspective.”

“Efficient Fe leaching under cooler conditions might have been related to elevated olivine weathering at cryogenic temperature. Freezing can increase the acidity of residual water, which counterintuitively accelerates the kinetics of chemical weathering...”

This is an interesting idea. I think it'd be helpful here to specify the pH ranges we're talking about, as acidic leaching is argued against as the main mechanism earlier in the text. Can the authors quantify the magnitude of the effect they're talking about here? Seems like it matters for the interpretation of their data.

Reply: We agree you very much. In this revised version, the magnitude of the effect was quantified, and the pH ranges were specified to address your concerns according to Nile's work¹. We added the point in lines 139-143: “Freezing can increase the acidity of residual water, potentially resulting in a pH level below 0, which counterintuitively accelerates the kinetics of chemical weathering to alter minerals at subfreezing temperatures^{2,3}. The weathering rate measured at -40°C is comparable to forsterite dissolution rates in a pH of 2.5 solution at 25°C¹.”

“Lower MAT can amplify the freezing, which can sustain chemical weathering and make the residual water more acidic (Figure 4a). As a potentially counterintuitive result, more Fe can be released under colder conditions”

This text is a bit repetitive with the sentence I just quoted above. I suggest editing to improve the flow.

Reply: We agree that the text was repetitive. We revised the sentence in lines 148-149 to improve the flow: “Under lower MAT, freezing can be amplified therefore can facilitate release of Fe and subsequent leaching.”

“. The relatively limited distribution of the permanent ice deposits may explain why glacial features are not extensively identified in the southern highlands”

It's worth citing and discussing some of the more recent literature on this topic here, e.g.:

Grau Galofre, A., Jellinek, A.M. and Osinski, G.R., 2020. Valley formation on early Mars by subglacial and fluvial erosion. *Nature Geoscience*, 13(10), pp.663-668.

Boatwright, B.D. and Head, J.W., 2021. A Noachian proglacial paleolake on Mars: fluvial activity and lake formation within a closed-source drainage basin crater and implications for early Mars climate. *The Planetary Science Journal*, 2(2), p.52.

Reply: Thank you for the suggestion to cite and discuss recent literature on this topic. The two references were incorporated in lines 178-180: “Above 3.0 km, subglacial erosion⁴ can form channels and proglacial fluvial channels can be formed below 3.0 km⁵ during the Noachian period”.

“Assuming the global average surface Fe abundance represents the original Fe abundance in the Noachian terrains”

I didn't quite follow the logic of this statement. The Fe content of e.g. Hesperian plains should be representative of the starting Noachian value only if they are sourced from mantle material with the same Fe content. Do we have independent constraints on this from mantle evolution modeling or other sources?

Reply: We use the Fe abundance of ALH84001, the only Martian meteorite with a Noachian age, as independent constraint for estimating the Fe abundance of Mars Noachian crust. As the Fe abundance of meteorite ALH84001 (~14.3 wt.%)^{6,7} is similar to the global surface average Fe abundance bases on GRS result. It is reasonable to propose that 14.3 wt.% is a good estimate of the Fe abundance of Noachian crust. We made corresponding changes in lines 193-199: “Martian meteorite ALH84001 with an igneous age of 4.091 ± 0.030 billion years is the only meteorite, which can be used to constrain the composition of Martian Noachian crust. The Fe abundance of meteorite ALH84001 is ~14.3 wt.%^{6,7}, which is similar to the global surface average Fe abundance bases on GRS result. Assuming ~14.3 wt. % represents the original Fe abundance in the Noachian terrains, ~0.58 wt.%, ~0.15 wt.%, and ~0.10 wt.% Fe was leached during the Early, Middle, and Late Noachian epochs, respectively (Fig. 3b).”

"A recent modeling result by Fan et al., (2022) indicates that the greenhouse warming effect could play a more important role than atmospheric pressure in diabatic cooling" It's worth noting that the Fan et al. preprint mischaracterizes past work on this topic, as neither Wordsworth (2016) or the earlier Forget et al. (2013) claimed that sensible heat fluxes are solely responsible for adiabatic cooling of surfaces under a thicker atmosphere (see e.g., Wordsworth 2016; p. 396).

Reply: Thank you for the comment. Fan et al., (2022)'s finding was positively discussed and emphasized several times by reviewers 2&3. Wordsworth (2016)'s work is well cited and discussed in the previous paragraph before the one we are talking. In light of your comment and the suggestions from reviewers 2&3, we revised the sentence in lines 248-250 to moderate the tone of the statement and provide a more accurate representation of the cited works: “In addition to atmospheric pressure, recent results by Fan et al., (2023)⁸ and Kite et al., (2022)⁹ suggest that the

greenhouse warming effect could also play an important role in diabatic cooling.”

Fig. 1: The authors show sulfates on the plot but don't discuss them much in the text. Are their results consistent with observations of sulfur minerals on Noachian and Hesperian terrain?

Reply: Thank you for pointing out the lack of discussion on sulfates in the text. We recently published a study¹⁰ in Science Advances about the formation of sulfates by freeze-thaw cycle in low latitudes. We included a discussion about sulfates t sulfates in the final paragraph of main text in lines 284-301: “The seasonal freeze-thaw region has transitioned from high-elevation areas to high-latitude regions as the climate mode has shifted with the gradual loss of H and H₂ in the atmosphere. With further cooling of Mars after NHT, seasonal freeze-thaw cycles could have migrated to low-latitude region during the Hesperian period. Owing to drying of Noachian Mars by sequestration of water in the crust¹¹, Hesperian Mars was arid with much less top down-leaching. Cryosuction during freezing can induce accumulation of water and ions from groundwater in the near-surface environments¹². During the thawing stage, strong evaporation under oxidizing conditions during the Hesperian period, can precipitate salts and ferric oxides⁶⁷. The enrichment of sulfates at Meridiani, the landing site of the Opportunity rover, can be well explained by seasonal freeze-thaw cycles¹⁰ according to the geochemical trends and textural observations. The evidence includes upwards enrichment of S but depletion of Cl, the upper bleaching and brecciated zone and lower darker and less brecciated zone with a sudden horizontal boundary in a meter-scale section, and the platy structure with abundant coarse planar voids but little fine voids. During the Hesperian period, the intersecting regions with high surface temperature (low latitude region) and a shallow water table promoted seasonal freeze-thaw with upward migration of groundwater via cryosuction, thus providing favorable conditions for the formation of sulfates¹⁰. This mechanism may explain why sulfates (blue squares in Fig. 1b) tend to be enriched in low latitude regions.”

Fig. 4: I strongly suggest changing “reducing greenhouse warming” to “decreasing greenhouse warming” here to avoid confusion.

Reply: Thank you for the suggestion. To avoid any confusion, we deleted the phrase.

Reviewers 2 and 3 co-reviewed and have submitted identical reports.

Reviewer 2 and 3

The paper analyzes remote sensing data to explore surface Fe abundance on ancient Martian terrains from the early Noachian to the Noachian-Hesperian Transition epoch. The authors find a gradual decrease of Fe depletion, as well as a transition of spatial pattern from altitude-dominated to latitude-dominated over time. The authors conclude that the Fe transition can be explained by the change of iron leaching intensity under a reducing environment, which is ultimately connected to a transition of atmospheric redox on early Mars. This paper provides new geochemical evidence relevant to the climate change on early Mars. It contributes to a converging view among the Mars community, together with other data papers and modeling papers. However, as a climate modeler, I do have several major concerns on data analysis and literature review. I believe the paper will be much more convincing to climate scientists once the comments are resolved. Since these comments are straightforward to apply, I recommend minor revisions to the editor.

Major comments:

1) *The citations to climate papers needs to be improved.* For example, the authors state that the transition from icy highlands to the modern mode is not fully understood at Line 29-31, with the refence to James Head’s conference talk. Yet the transition and mechanism were discussed in depth in Kite et al. (2022) and Fan et al. (2023). The other significant example is the citation to Wordsworth (2017) at Line 38. Yet Wordsworth (2017) is neither the earliest, nor the most precise paper on the greenhouse effect caused by a reducing atmosphere. Examples like these are numerous throughout the Introduction and Discussion. My suggestions to address this issue are listed below, with the reference list attached in the end.

Reply: We appreciate your thorough review and suggestions regarding the citations to climate papers in our manuscript. We revised the Introduction and Discussion to better incorporate the relevant literature, including Kite et al. (2022) and Fan et al. (2023). In lines 29-34, we now discuss the potential mechanisms and timing of the climate mode transition: “Several potential mechanisms¹³ have been proposed to drive the climate mode change, including the decrease in atmospheric CO₂^{3,14} or atmospheric H₂^{8,9}, or a reduction in cloud radiative effect^{8,9,15}. However, the exact process behind the climate mode change remains unclear. While significant cooling⁹ and drying¹¹ have been suggested from Noachian Mars to Hesperian Mars, the timing of this climate mode transition remains poorly constrained¹⁶.”

We made any necessary adjustments to other citations throughout the manuscript, following your suggestions. We believe these changes address your concerns and improve the overall quality of our discussion on climate research.

Line 29: Early modeling results should cite Wordsworth (2016).

Reply: Done.

Line 29-31: The transition has been discussed in Kite et al. (2022) and Fan et al. (2023). There is not yet a consensus on which mechanism drives the change of greenhouse effect. It could be the decrease of CO₂ (Wordsworth, 2016; Kite, 2019), or the decrease of H₂ (Fan et al., 2023), or cloud radiative effect (Urata & Toon, 2013).

Reply: As mentioned, we added the two sentences here to better reflect previous work and lack of consensus on the driving mechanism in lines 29-34: “Several potential mechanisms¹³ have been proposed to drive the climate mode change, including the decrease in atmospheric CO₂^{3,14} or atmospheric H₂^{8,9}, or a reduction in cloud radiative effect^{8,9,15}. However, the exact process behind the climate mode change remains unclear. While significant cooling⁹ and drying¹¹ have been suggested from Noachian Mars to Hesperian Mars, the timing of this climate mode transition remains poorly constrained¹⁶.”

Line 32-34: There are other papers that suggest the presence of reducing gases based on the modern record. Examples include, for atmospheric evolution, Yoshida & Kuramoto (2020) and Wordsworth et al. (2021), and for surface river records, Kite et al. (2022).

Reply: We cited these references according to your suggestions.

Line 34-36: Please add the references that quantified the potential source of H₂:

Volcanos: Ramirez et al. (2014); Batalha et al. (2015)

Serpentinization: Liu et al. (2021a, 2021b), Tutolo & Tosca (2023)

Impacts: Haberle et al. (2019)

Reply: Thank you for bringing these references to our attention. We added them at corresponding places in this version.

Line 35-37: The reference to Tutolo & Tosca (2023) should be merged to serpentization papers at Line 34-36. If the authors do think this paper needs more exposure, I recommend switching the sentence to Discussion.

Reply: We agree with you. We cited Tutolo & Tosca (2023) for serpentization. Also, the sentence was switched to Discussion in lines 213-214: “Serpentization might have produced significantly more reducing gases than previously thought¹⁷ owing to the high iron content in the Martian crust¹⁸.”

Line 38: Using Wordsworth (2017) as the only one citation is unbalanced. The hydrogen greenhouse effect hypothesis can be traced at least as far back to Sagan (1977). The modern discussions were initiated by Wordsworth & Pierrehumbert (2013) and Ramirez et al. (2014). The most up-to-date data are Turbet et al. (2020), Godin et al. (2020), Kamada et al. (2021, 2022).

Reply: Thank you for the comment and suggestion. We added other references recommended by you here.

Line 39-41: If the authors want to keep reference 7-9, Kamada et al. (2021, 2022) should be added for their contribution in quantifying the greenhouse effect in GCMs.

Reply: Done.

Line 109-110: Cite Wordsworth (2016)

Reply: Done.

Line 199-200: Cite Kite (2022)

Reply: Done.

Line 219: Cite Kite (2022)

Reply: Done.

2) *Data analysis for the altitude dependence should be improved.* The term “icy highlands” can be confusing because under the “icy highlands” regime (or equivalently the strong greenhouse regime), both high altitude and high latitude are cold traps of a planet.

Reply: We agree that.

Thank you for your comment regarding the term “icy highlands” and its potential for confusion. We have revised the text in lines 27-29 to better describe the temperature and water ice distribution during the strong greenhouse regime on early Mars.

“Conversely, modeling results suggest that temperature was predominantly elevation-

dominant and water ice mainly accumulated in highlands and polar regions regardless of the mean annual temperature (MAT) on early Mars¹⁹⁻²¹.”

For both endmembers of the temperature pattern, a large equator-to-pole temperature gradient ($dT > 30$ K) on early Mars is expected (see Figure 1 in Fan et al., 2023 for the example). If the authors plot all data points on a temperature-altitude plot, the correlation coefficient can be much smaller than it should be for a fixed solar heating due to the equator-to-pole insolation gradient. This issue can be much worse for scenarios with a small obliquity (this may be why Wordsworth, 2016 only show the case with obliquity = 41.8 deg). My suggestion is to decompose the altitude effect and latitude effect for each epoch. For example, the authors could use three latitudinal bands (tropics, northern high latitudes, southern high latitudes) like Fan et al. (2023), or even more latitudinal bands (from low to high latitudes) like Faulk et al. (2017). I expect the authors will get higher coefficients for the altitude dependence during the early Noachian and Middle Noachian epoch. Even if the coefficients are still low, it offers more information, specifically that the insolation at one latitudinal band was disrupted by orbital variations. For the latitudinal dependence, the authors may choose not to decompose the NHT or even the late Noachian epoch due to the lack of sufficient data. It is OK because the temperature pattern for the modern regime is 1-D. But again, it is expected that for early epochs the latitudinal coefficients can be larger once the altitudinal dependence is excluded.

Reply: Thank you for your insightful suggestions on improving the data analysis for the altitude dependence. We followed your advice and decomposed the latitude to three latitudinal bands ($>20^{\circ}\text{N}$, $20^{\circ}\text{S}-20^{\circ}\text{N}$, and $<20^{\circ}\text{S}$) for Early and Middle Noachian in this version (Supplementary Fig. 4). Late Noachian and the NHT were not decomposed due to the lack of sufficient data as suggested. As you expected, we get high coefficients for tropics and northern high latitudes. For southern high latitudes ($>20^{\circ}\text{N}$), the trend of Fe abundance with elevation is not clear, because limited top-down leaching hence relatively high Fe abundance in permafrost region with high elevation. We made revision correspondingly in the text in lines 97-102: “To be more specific, the decreasing trend is pronounced in northern high latitudes ($>20^{\circ}\text{N}$, Supplementary Fig. 4a and 5) and tropics ($20^{\circ}\text{S}-20^{\circ}\text{N}$, Supplementary Fig. 4c and 5), both of which have elevations lower than 3 km. For southern high latitudes ($>20^{\circ}\text{N}$), the average Fe abundance of regions with elevation above 3.0 km (~ 14.0 wt.%) is higher than that of regions between 0-3.0 km (~ 13.0 wt.%) (Supplementary Fig. 4b and 5).” and in lines 170-175: “Most regions with elevation above 3.0 km are in southern high-latitude region (Supplementary Fig. 5). The limited top-down leaching in high-elevation permafrost regions but efficient Fe leaching in lower-

elevation regions can account for the absence of a decreasing trend of Fe abundance with elevation in the southern high latitudes (Supplementary Fig. 4). It is also possible that obliquity variations, leading to the insolation/temperature variations with latitude, can disrupt the Fe abundance distribution with elevation.” and in lines 318-320: “Three latitudinal bands ($>20^{\circ}\text{N}$, $20^{\circ}\text{S}-20^{\circ}\text{N}$, $<20^{\circ}\text{S}$) was further divided to examine the correlation between Fe abundance and elevation during the Early and Middle Noachian epochs to reduce the latitude effect (Supplementary Figs. 4 and 5).”

Decomposing will make the manuscript's figures more complicated (e.g., Fig. S1). To keep the figures in the main text simple, possible modifications include (1) choosing a typical latitudinal band for the early epoch in Figure 2 and (2) replacing one green square to multiple green squares in Figure 3b.

Reply: Thank you very much for the suggestions. For suggestion (1) we decompose latitude for the Early and Middle Noachian terrains to show the relationship between Fe abundance and elevation. The figures were added in Supplementary Fig. 4. We keep original Figure 2b in main text mainly because including all latitudes can better show the transition from permafrost (>3 km), seasonal freeze-thaw ($-3\text{km}-3\text{km}$), and no freeze-thaw ($< 3\text{km}$) at different elevations. But the three processes cannot be indicated in any single latitudinal band together. For suggestion (2), multiple green hollow squares from three latitude were added in Figure 3b.

Other comments:

Title: Please revise as "... on Noachian Early Mars" or "on Early Mars between 4.1 Ga – 3.7 Ga", for precision.

Reply: We appreciate your recommendation for improving the precision of the title. In accordance with your suggestion, "Noachian" was added to the title to specify the period being discussed in the study. The revised title of the manuscript is now: "Atmospheric oxidation drove climate change on Noachian Mars".

Abstract: Overall the statements are too strong. At the current stage of this paper, the transition of altitude-dependence needs to be further verified (see major comment #2); the role of igneous processes in changing iron abundance needs further work; the climate & redox interpretation to the temperature pattern is complex (see the comments on Discussion). With those uncertainties, the story implied from the work is still a hypothesis, not a fact. Recommend revisions include (1) changing "imply" at Line 12 to "suggest"; (2) changing the last sentence as "The change of iron abundance suggest ..." or "We think ...".

Reply: We agree you with the comment and suggestions on the abstract.

Modifications were made according to suggestions in lines 14-16 "These observations suggest: (a) low-temperature conditions contribute to surface iron depletion, likely facilitated by anoxic leaching through freeze-thaw cycles under a reducing atmosphere" and in lines 19-20: "We think atmospheric oxidation led to Mars becoming cold and bipolar in its early history."

Line 46-47: Please justify the focus on the Noachian terrains for the later part of the paper (i.e., please explain why the conclusions cannot be extrapolated to Hesperian and Amazonian).

Reply: We justify the focus on the Noachian terrains by rearranging the order of sentences and add two new sentences in lines 52-58: "Fe mobility can be affected by acidity, aqueous chemistry, redox state, and temperature during aqueous alteration²²⁻²⁵. Therefore, the redox and climate transitions could have been captured by the distribution of Fe abundance at the surface of Mars, when there is communication between the atmosphere, hydrosphere and lithosphere was efficient and top-down leaching during the Noachian period. The characteristics of surface water can be recorded through Fe mobility, however, the Hesperian and Amazonian terrains do not exhibit signs of Fe depletion (Fig. 1a). Consequently, the study predominantly focuses on investigating the Noachian terrains. "

Line 51-58: A number of unpreferred interpretations are explained in this paragraph, but somehow the preferred one is not. It is surprising that “Fe leaching under a reducing condition” is explained late in Discussion. Since it is the key concept of the paper, please give a basic introduction here (with one or two sentences).

Reply: Thank you for your suggestion to include a brief introduction to the concept of "Fe leaching under a reducing condition" earlier in the manuscript. We agree that this is a key concept of the paper and should be mentioned earlier. In response to your recommendation, we added two sentences to explain the mobility of Fe under different redox conditions in lines 65-67. “On the other hand, the mobility of Fe can be strongly affected by water redox. Fe is immobile and tend to precipitate as Fe (oxyhydro)oxides under oxidizing conditions but becomes soluble and mobile as Fe(II) under reducing conditions.”

Line 60-67: The reasoning here may be hard to follow for some readers. What does the discrepancy from CRISM data say? Does that mean another mechanism for iron depletion? Why in the end the authors go back to the same conclusion of chemical weathering?

Reply: Thank you for pointing out the potential confusion in the reasoning presented. In response, we stated that anoxic chemical weathering and leaching can happen with the formation of clay minerals under temperature above freezing or without the formation of clay minerals under temperature around or below 0°C. Here, we would like to emphasize that anoxic chemical weathering and leaching under temperature around or below 0°C account for the Fe leaching rather than that happened under temperature well above freezing. We clarified the statement in lines 69-78: “Anoxic chemical weathering and leaching can happen with the formation of clay minerals under temperature well above freezing¹¹ or without the formation of clay minerals under temperature around or just above 0°C³⁷. Although overlaps existed, there is no clear correlation between the distributions of Fe depletion zones (dark red area in Fig. 1b) identified through GRS data and paleosols rich in clay minerals (pink dots in Fig. 1b) detected via infrared remote sensing using Compact Reconnaissance Imaging Spectrometer for Mars (CRISM) data (Fig. 1b). The decouple is possibly related to the limited clay minerals’ detections by lack of areal coverage of CRISM data compared to the global coverage and larger footprint of GRS data. But it is more likely that the Fe depletion was caused by reducing chemical weathering and leaching under temperatures around or just above 0°C, which could inhibit the formation of clay minerals²⁶.” We believe this revision provides a clearer explanation, making the manuscript more coherent and easier to understand for readers.

Line 60-63 & Figure 1:

(1) Recommend revising the label & legend in Figure 1b alongside the text. Which part of Figure 1b is referred to exactly? Is it comparing the iron depletion zone (dark red?) and paleosols (brown? pink?)?

Reply: Yes, we are comparing the iron depletion zone and the paleosols. We revised the legend of the Figure in lines 507-509: “Spatial distribution of clays-rich paleosols²⁷ (pink cycles), sulfates²⁸ (blue squares), and Fe depletion region²⁹ (dark red area) on a Mars global map²⁹”. Corresponding modification was also made in main text in lines 71-74: “Although overlaps existed, there is no clear correlation between the distributions of Fe depletion zones (dark red area in Fig. 1b) identified through GRS data and paleosols rich in clay minerals (pink cycles in Fig. 1b) detected via infrared remote sensing using Compact Reconnaissance Imaging Spectrometer for Mars (CRISM) data (Fig. 1b).”

Why shows sulfate?

Reply: It is because sulfate might be products of seasonal freeze-thaw cycles according to our recent publication¹⁰. To explain more about it, a new paragraph was added in lines 284-301: “The seasonal freeze-thaw region has transitioned from high-elevation areas to high-latitude regions as the climate mode has shifted with the gradual loss of H and H₂ in the atmosphere. With further cooling of Mars after NHT, seasonal freeze-thaw cycles could have migrated to low-latitude region during the Hesperian period. Owing to drying of Noachian Mars by sequestration of water in the crust¹¹, Hesperian Mars was arid with much less top down-leaching. Cryosuction during freezing can induce accumulation of water and ions from groundwater in the near-surface environments¹². During the thawing stage, strong evaporation under oxidizing conditions during the Hesperian period, can precipitate salts and ferric oxides⁶⁷. The enrichment of sulfates at Meridiani, the landing site of the Opportunity rover, can be well explained by seasonal freeze-thaw cycles¹⁰ according to the geochemical trends and textural observations. The evidence includes upwards enrichment of S but depletion of Cl, the upper bleaching and brecciated zone and lower darker and less brecciated zone with a sudden horizontal boundary in a meter-scale section, and the platy structure with abundant coarse planar voids but little fine voids. During the Hesperian period, the intersecting regions with high surface temperature (low latitude region) and a shallow water table promoted seasonal freeze-thaw with upward migration of groundwater via cryosuction, thus providing favorable conditions for the formation of sulfates¹⁰. This mechanism may explain why sulfates (blue squares in Fig. 1b) tend to be enriched in low latitude regions.”

(2) Please include a geological map with different epochs of Noachian in the Supplementary Material.

Reply: It is a good suggestion. We added a geological map (Supplementary Fig. 1) with different epochs of Noachian in the Supplementary Material as suggested.

Line 66: How low does the temperature need to be? Specifying the temperature range required here will be very helpful to quantifying the atmospheric concentration / greenhouse effect for future climate modeling works.

Reply: The temperature can be around or just above 0°C. We added the temperature in lines 79-81: “But it is more likely that the Fe depletion was caused by reducing chemical weathering and leaching under temperatures around or just above 0°C, which could inhibit the formation of clay minerals²⁶.”

Line 77: Please give a basic description of the research approach first. Some readers might not know where “bins” come from.

Reply: Thank you for the suggestion. A basic description of the research methods was added in lines 92-93: “The global geologic map of Mars²⁹ (Supplementary Fig. 1) was used to identify the apparent relative surface age of each 5° × 5° bin of the GRS data³⁰.”

Line 77-79: This goal of this paragraph may be confusing to some readers. Is it necessary to include it in the main text (rather than in the Methods)?

Reply: The short sentence was deleted in this version.

Figure 2:

1. Very interesting results! Could the authors please add more interpretation why the snowline can explain the transition of trend? Fan et al. (2023) predicts either a temperature discontinuity for the weak greenhouse case, or a smooth trend following an adiabat for the strong greenhouse case. Yet the data seems not applicable to either case. Why should data above snowline should have the average value? If chemical weathering does not take place under this regime, then should it not record the largest iron abundance? Please can the authors elaborate and/or explain?

Reply: Thank you for your interest in our results and for raising these questions. We think the distribution trend of Fe abundance is consistent with a smooth trend following an adiabat for the strong greenhouse case. In response to your concerns, we have made the following revisions to our manuscript:

- 1) We modified the term "snowline" to "permafrost".
- 2) We explained that chemical weathering can indeed take place in permafrost, but Fe

leaching cannot occur because surface water cannot leach downwards when permafrost is present. As a result, the Fe abundance in the highest-elevation region is close to the original Fe abundance in the Noachian crust, which may be 14.3 wt. %.

3) We also added that the top-down increasing trend of Fe abundance can be contributed by the precipitation of leached Fe accumulated in the low elevation by intermittent oxidizing atmosphere. Consequently, the lower-elevation region, rather than the highest-elevation region without leaching, should record the largest iron abundance. This statement was added in lines 152-154.

We believe these revisions provide a clearer interpretation of the transition of the trend and its connection to the permafrost, addressing the concerns raised and improving the overall understanding of our results.

2. Please can the exclusion of the -3.5 line be justified? For the freeze-thaw cycle, alternatively you can say the cycle extends to -4~-3 km and the transition at bottom is just a lack of sufficient data. If the authors wish to keep their current boundary of no freeze-thaw, it is recommended that they give additional discussion on how clay formation explains the iron abundance at low altitudes.

Thanks for the suggestion. We added discussion to explain the iron abundance at low elevations in lines 154-159: “On the other hand, under temperature above freezing, chemical weathering can lead to direct top-down leaching (Fig. 4c). The rate of chemical reaction and Fe release and leaching is positively related to temperature, therefore more Fe can be depleted with decreasing elevation below -3 km, as indicated in Fig. 2a. The transition in chemical weathering rate and leaching efficiency partly account for the broad distribution range of Fe abundance at elevations between -3 km and -4 km (Fig. 2a).”

3. Please add the timing for each panel (early & middle □when? late and NHT □when?)

Reply: Thank you for your suggestion to add the timing for each panel in our study. We understand that providing the timing would make the presentation of our results more direct. However, there is currently no consensus on the boundary between the Noachian and Hesperian epochs (with estimates ranging from ~3.7 Ga to ~3.5 Ga). As a result, the timing for each epoch (early, middle, and late) is not well-defined. Given the uncertainty surrounding these age estimates, we have decided not to include specific timings for each epoch in our manuscript. We believe that the relative age information provided by the terms Early, Middle, and Late Noachian is sufficient to convey the necessary age information without introducing potential

misunderstandings due to uncertain age boundaries.

4. How does the figure look like when the altitude and latitude effect are fully decomposed? (see major comment #2). For example, how does latitude relation work under the early time? How does the elevation relation work under the late time? The authors might want to put most of the results into supplementary information...

Reply: We decomposed latitude for the Early and Middle Noachian terrains to show the relationship between Fe abundance and elevation. The figures were added in Supplementary Fig. 4. Late Noachian and the NHT were not decomposed due to the lack of sufficient data as suggested. As you expected, we get high coefficients for tropics and northern high latitudes. For southern high latitudes (>20°N), the trend of Fe abundance with elevation is not clear, because limited top-down leaching hence relatively high Fe abundance in permafrost region with high elevation.

5. Please add $n = ?$ for the number of the data included.

Reply: Done.

Figure S1:

Recommend add timing (following Figure 2)

Reply: Thank you for your suggestion to add the timing for each panel in our study. We understand that providing the timing would make the presentation of our results more direct. However, there is currently no consensus on the boundary between the Noachian and Hesperian epochs (with estimates ranging from ~3.7 Ga to ~3.5 Ga). As a result, the timing for each epoch (early, middle, and late) is not well-defined. Given the uncertainty surrounding these age estimates, we have decided not to include specific timings for each epoch in our manuscript. We believe that the relative age information provided by the terms Early, Middle, and Late Noachian is sufficient to convey the necessary age information without introducing potential misunderstandings due to uncertain age boundaries.

Recommend decomposing the latitude and altitude effect (following Figure 2)

Figure 3a:

1. If my reading is correct, the difference between Early Noachian average (red dot) and global average (horizontal dashed line), which is 0.58, is not about four times larger compared to the difference between Middle Noachian average and global average, which is 0.15. Is there something wrong with data visualization?

Thank you for the concern. The abundance of Fe leaching in Early Noachian epoch should be the difference between Early Noachian average and Middle Noachian

average, which should be 0.58 wt.%. Similarly, the 0.15 wt.% is the difference between Middle Noachian average and Late Noachian average. The abundance of Fe leaching in Early Noachian epoch (0.10 wt.%) is calculated between Late Noachian average and global average. We clarified that in lines 114-117 “Their values increase gradually with increasing amounts at ~0.58 wt.% from the Early to the Middle Noachian terrains, ~0.15 wt.% from the Middle to the Late Noachian terrains, and ~0.10 wt.% from the Late Noachian terrains to the global surface average, respectively (Fig. 3b).” and in lines 541-543: “The abundance of Fe leaching in each epoch is calculated based on the value of Fe depletion relative to adjacent unit (for Early Noachian and Middle Noachian) or global surface average (for Late Noachian).”

2. How reducing would the atmosphere be (e.g., partial pressure of H₂) for iron depletion? Please add a citation (or calculation).

Reply: Thank you for your question regarding the partial pressure of H₂ in the atmosphere and its relevance for iron depletion. We acknowledge that the partial pressure of H₂ is an important factor for climate modeling. However, determining the partial pressure of H₂ in the atmosphere based solely on Fe depletion values is not feasible in our study. To calculate the partial pressure of H₂, a top-down Fe distribution paleosol profile with known pH and temperature would be required. Unfortunately, we do not have all the necessary constraints at this time, which prevents us from accurately calculating the partial pressure of H₂ in the atmosphere.

3. The light blue numbers and horizontal lines in Figure 3a are confusing and distracting. Since there is already a clear presentation of them in Figure 3b, I would recommend just removing them in Figure 3a.

Reply: We agree with the comment. The lines in Figure 3a were removed in this version.

Figure 3b: Please add error bars for the correlation analysis (same in Figure S1).

Reply: Thank you for your suggestion to add error bars for the correlation analysis in Figures 2 and 3b and Supplementary Figures 4 and 5. We appreciate the importance of providing uncertainty information for our results. In response to your recommendation, we have added error bars to both figures, representing the 95% confidence intervals of the correlation coefficients (r). We performed the correlation analysis using the `cor.test()` function in R, with a sufficient number of data points to ensure the statistical robustness of our results. We believe that the inclusion of these error bars will provide a more complete and accurate representation of our findings,

enhancing the overall quality of our manuscript.

Tables:

1. Decompose the latitude and altitude effect (following Figure 2)

Reply: Done, the analyzing results for different latitude bands ($>20^{\circ}\text{N}$, 20°S - 20°N , $<20^{\circ}\text{S}$) can be found in Supplementary Tables 2-4).

2. A missing “e” for “Latitud” in Table S2.

Reply: We modified the typo.

Line 122: There is no direct demonstration of infiltration or Fe leaching in Figure 3b (referring to somewhere else?).

Reply: We removed the citation of Figure 3b here in the revised version our manuscript.

Line 123-124: The necessity of a reducing environment has not been explicitly stated yet (“mobile iron = reducing” is unclear for atmospheric scientists).

Reply: Thank you for pointing out the unclear statement regarding the necessity of a reducing environment in Lines 123-124. We understand the importance of providing a clear explanation, especially for readers from different scientific backgrounds. To address your concern, we have included an explanation in the Introduction (Lines 63-65) of the revised manuscript in lines 65-67: “On the other hand, water redox can greatly influence Fe mobility. Specifically, Fe is immobile and tend to precipitate as Fe (oxyhydro)oxides under oxidizing conditions but becomes soluble and mobile as Fe(II) under reducing conditions.” In addition, we added one sentence in lines 145-146: “Under reducing conditions, Fe is dissoluble and mobile as Fe(II).” We believe that these revisions will improve the clarity of our manuscript for all readers, including atmospheric scientists.

Fig. 4:

1. Based on the reasoning in this Figure, the text in Figure 2a should be revised from “snow line” to “permafrost”.

Reply: Thank you for pointing out the inconsistency in terminology between the text and Figure 2a. “Snow line” was revised to “permafrost” in Figure 2a in this version.

2. Following the statement from Line 136-140, iron abundance stays high under a clay-forming regional climate. Inconsistently, all the strange signals (wide data range for $-4\sim-3$ km & a transition of trend for $-4\sim-5$ km) are attributed to clay mineral

formation. It remains unclear to me how the two stories can be reconciled.

Reply: Thank you for pointing out the inconsistency in our explanation of iron abundance and clay mineral formation in relation to elevation. We appreciate your attention to this matter, as it is crucial to provide a clear and coherent narrative for our readers. To address your concern, we have modified the text in lines 154-159: “On the other hand, under temperature without freezing, chemical weathering can form clay minerals with direct top-down leaching (Fig. 4c). The rate of chemical reaction and Fe leaching is positively related to temperature, therefore Fe is depleted with decreasing elevation, as indicated in Fig. 2a. The transition in chemical weathering rate and leaching efficiency partly contribute to the broad distribution range of Fe abundance at elevations between -3 and -4 km.” Hope the modification make the logic clearer.

Line 182 – 186: The iron enrichment during the late stage is important. Maybe move to Results?

Reply: We agree with the comment. The iron enrichment during the Noachian-Hesperian Transition was presented in the Results section in lines 117-120: “However, the Fe abundance at the surface of the NHT terrains is remarkably high (~15.4 wt.%), which is ~1.2 wt.% higher than that of the Late Noachian terrains and 1.1 wt.% higher than the global surface average (Fig. 3a).”

Line 200-202: I don’t think this is true. A transition from ADD to LDD would imply a net decrease of the greenhouse effect, which means a lower MAT.

Reply: We agree the comment. We removed the sentence in this version.

Line 212-239: Even taking the three aspects of iron abundance evolution to be true, the interpretations to climate & redox can be more complex than what the authors have shown. The less iron depletion over time can be caused by a warmer global mean temperature, or a less reducing environment. The more latitudinal dependence can be caused by a thinner atmosphere, or a lower obliquity, or less H₂ greenhouse effect, or less CO₂ greenhouse effect. The less altitude dependence can be caused by less H₂ greenhouse effect or less CO₂. Here red means a warming climate, cyan means a slightly colder climate, and blue means a much colder climate. The overlap of the interpretations on the three aspects is the conclusion of this paper, but the complexities/uncertainties for this logical step needs to be demonstrated.

Reply: We appreciate your thoughtful comments on the potential complexities and uncertainties in interpreting the three aspects of iron abundance evolution in relation to climate and redox conditions. Your suggestions have prompted us to expand our discussion and consider alternative explanations for the observed trends in our data. In

response to your recommendations, we have added a more thorough discussion in Lines 256-266: “Figure 3b show that, from the Early to Late Noachian epoch, Fe leaching intensity decreases, covarying with coefficients of Fe abundance with latitude and elevation (Fig. 3b). The reduced Fe depletion (Fig. 3a) over time during the Noachian period can be caused by a warmer global mean temperature (Fig. 4), or a less reducing environment. The increased latitude dependence can be caused by a reduced H₂ greenhouse effect or a reduced CO₂ greenhouse effect (a thinner atmosphere)^{21,31}, or a lower obliquity. The decreased elevation dependence can be caused by a reduced H₂ greenhouse effect⁹ or less CO₂⁸. The overlapping interpretation of these three aspects is the decrease in the amount of atmospheric H₂, over time during the Noachian period. In other words, it is highly possible that the Martian climate mode transition from EDD to LDD during the Noachian (Fig. 3b) was mainly driven by the atmospheric redox transition.”

References:

Kite, E. S., Mischna, M. A., Fan, B., Morgan, A. M., Wilson, S. A., & Richardson, M. I. (2022). Changing spatial distribution of water flow charts major change in Mars's greenhouse effect. *Science Advances*, 8(21), eabo5894.

- Fan, B., Jansen, M. F., Mischna, M. A., & Kite, E. S. (2023). Why are Mountaintops Cold? The Transition of Surface Lapse Rate on Dry Planets. *arXiv preprint arXiv:2311.10151*.
- Wordsworth, R., Kalugina, Y., Lokshtanov, S., Vigasin, A., Ehlmann, B., Head, J., ... & Wang, H. (2017). Transient reducing greenhouse warming on early Mars. *Geophysical Research Letters*, *44*(2), 665-671.
- Wordsworth, R. D. (2016). The climate of early Mars. *Annual Review of Earth and Planetary Sciences*, *44*, 381-408.
- Urata, R. A., & Toon, O. B. (2013). Simulations of the martian hydrologic cycle with a general circulation model: Implications for the ancient martian climate. *Icarus*, *226*(1), 229-250.
- Kite, E. S., Steele, L. J., Mischna, M. A., & Richardson, M. I. (2021). Warm early Mars surface enabled by high-altitude water ice clouds. *Proceedings of the National Academy of Sciences*, *118*(18), e2101959118.
- Yoshida, T., & Kuramoto, K. (2020). Sluggish hydrodynamic escape of early Martian atmosphere with reduced chemical compositions. *Icarus*, *345*, 113740.
- Wordsworth, R., Knoll, A. H., Hurowitz, J., Baum, M., Ehlmann, B. L., Head, J. W., & Steakley, K. (2021). A coupled model of episodic warming, oxidation and geochemical transitions on early Mars. *Nature Geoscience*, *14*(3), 127-132.
- Ramirez, R. M., Kopparapu, R., Zuger, M. E., Robinson, T. D., Freedman, R., & Kasting, J. F. (2014). Warming early Mars with CO₂ and H₂. *Nature Geoscience*, *7*(1), 59-63.
- Batalha, N., Domagal-Goldman, S. D., Ramirez, R., & Kasting, J. F. (2015). Testing the early Mars H₂-CO₂ greenhouse hypothesis with a 1-D photochemical model. *Icarus*, *258*, 337-349.
- Haberle, R. M., Zahnle, K., Barlow, N. G., & Steakley, K. E. (2019). Impact degassing of H₂ on early Mars and its effect on the climate system. *Geophysical Research Letters*, *46*(22), 13355-13362.
- Liu, J., Michalski, J. R., & Zhou, M. F. (2021). Intense subaerial weathering of eolian sediments in Gale crater, Mars. *Science Advances*, *7*(32), eabh2687.
- Liu, J., Michalski, J. R., Tan, W., He, H., Ye, B., & Xiao, L. (2021). Anoxic chemical weathering under a reducing greenhouse on early Mars. *Nature Astronomy*, *5*(5), 503-509.
- Tutolo, B. M., & Tosca, N. J. (2023). Observational constraints on the process and products of Martian serpentinization. *Science Advances*, *9*(5), eadd8472.

SAGAN, C. (1977). Reducing greenhouses and the temperature history of Earth and Mars. *Nature*, 269(5625), 224-226.

Wordsworth, R., & Pierrehumbert, R. (2013). Hydrogen-nitrogen greenhouse warming in Earth's early atmosphere. *science*, 339(6115), 64-67.

Turbet, M., Boulet, C., & Karman, T. (2020). Measurements and semi-empirical calculations of CO₂+ CH₄ and CO₂+ H₂ collision-induced absorption across a wide range of wavelengths and temperatures. Application for the prediction of early Mars surface temperature. *Icarus*, 346, 113762.

Godin, P. J., Ramirez, R. M., Campbell, C. L., Wizenberg, T., Nguyen, T. G., Strong, K., & Moores, J. E. (2020). Collision-induced absorption of CH₄-CO₂ and H₂-CO₂ complexes and their effect on the ancient Martian atmosphere. *Journal of Geophysical Research: Planets*, 125(12), e2019JE006357.

Kamada, A., Kuroda, T., Kasaba, Y., Terada, N., & Nakagawa, H. (2021). Global climate and river transport simulations of early Mars around the Noachian and Hesperian boundary. *Icarus*, 368, 114618.

Kamada, A., Kuroda, T., Kodama, T., Kasaba, Y., & Terada, N. (2022). Evolution of ice sheets on early Mars with subglacial river systems. *Icarus*, 385, 115117.

Reviewer 4

Dear Jiacheng Liu et al.,

This was an exciting manuscript to read and some big impacts for the planetary community. Below I have provided a major comment with some more modest suggestions to help strengthen the manuscript. Please reach out if you need any clarification and good luck with the process. Sincerely,

Mike Thorpe (michael.t.thorpe@nasa.gov)

Reply: Hello Mike, thank you for taking the time to review our manuscript and providing valuable feedback. We are glad that you found the manuscript exciting and impactful for the planetary community. We appreciate the suggestions you have provided and have addressed them in a point-by-point manner.

Major Comment:

The one major comment I have on this manuscript is to at least address or link some of these results to rover findings. As it is currently written, the processes seems very linear from a reducing to an oxidizing environment. But in Gale crater, we have examples within the stratigraphy that reduced phases and oxidized phases are fluctuating in a way that might be challenging to describe the paleoenvironment as a linear transition. Even more interesting is that we have reduced carbonates in the most recent drill holes, presumably highest in the stratigraphy. I feel this manuscript should address in situ measurements to strengthen any implications of this work.

Reply: Thanks for the comment and helpful information. We agree that it is important to consider the in situ measurements, as they can provide valuable context. Based on the comment, a new paragraph were added in lines 221-228: “These observations suggest a global and broad trend of oxidation, but it does not necessarily imply a linear transition from a reducing to an oxidizing environment. In fact, most of the reducing gases outgassing events (e.g., volcanoes and impact) are intermittent, which more likely contribute to an episodic reducing atmosphere and warm climate according to modeling result³². Also, the episodic reducing atmosphere and warm climate are more consistent with wet-dry cycling³³ and intermittent appearance of ferrous hydrous minerals^{34,35} at Gale crater identified by the Curiosity rover, as well as short-term chemical weathering events under warm conditions³⁶.”

In addition, we discussed the in situ geochemical and mineralogical observations from the Opportunity at Meridiani in the final graph of the manuscript, particularly in lines 292-297: “The enrichment of sulfates at Meridiani, the landing site of the Opportunity rover, can be well explained by seasonal freeze-thaw cycles¹⁰ according to the geochemical trends (Upwards enrichment of S but depletion of Cl) and textural observations (such as the upper bleaching and brecciated zone and lower darker and

less brecciated zone with sudden horizontal boundary in a meter-scale section, and the platy structure with abundant coarse planar voids but little fine voids).”

Modest Comments/Suggestions:

Can you discuss outliers a bit and how statistically meaningful these trends are. It's a r max of 0.47 for elevation and 0.77 for latitude but that's also with far fewer points.

Reply: Thank you for pointing out that. We talk more about the outliers in lines 229-236: “The observed negative correlation between Fe abundance and elevation during the Early and Middle Noachian epochs (Fig. 3b and Supplementary Fig. 3a) suggests that the temperature distribution was elevation dominant. The correlations are not very strong possibly due to the influence of bins with elevations above +3 km or below -3 km (Supplementary Fig. 3a) and the persistent effect of latitude on temperature and therefore Fe distribution, as well as the obliquity variations, during the Early and Middle Noachian epochs. Regardless of the relatively low r -value (Fig. 3b), the decreasing trend of r -value between Fe abundance and elevation over time makes sense.”

In line 89 the authors write “association between Fe abundance and latitude is weak.” These weak associations have r values not too far off from elevation. While I still think the trends and tightness of the grouping makes sense, it is something to note here.

Reply: Thank you for pointing out that. To describe the association more accurately, we reported the correlation coefficient directly in lines 108-112: “Instead, there is a decreasing trend in Fe abundance with increasing latitude (Fig. 2b), especially on southern Mars. The correlation coefficient between Fe abundance and latitude is not strong in the Early Noachian terrains, but strong and positive in the Late Noachian (+0.52) and Noachian-Hesperian transition (+0.77) terrains of southern Mars (Supplementary Fig. 3b). It means that the correlation coefficient (r) increases gradually over time in the Noachian (Fig. 3b).”

Line 126: The authors state a “negative correlation between Fe and Mg according to the global GRS” but only cite a reference. If this is a need to support the argument, this correlation should at least be portrayed in the supplementary. I think the authors are trying to suggest the closure effect and that since no other elements are increasing while Fe decreases, they need to support it with Mg rising. Also, in the lines above the authors address Fe vs. Mg olivine but another route for this trend could be that both phases are liberated during incipient alteration and then their behavior differs with increasing alteration. Could be another mechanism to address.

Reply: Thank you for the comment. After careful consideration, we decided to remove the sentence from our manuscript, as it is tangential to the main focus of our study. By doing so, we aim to maintain the clarity of our arguments.

Line 153: How warm?

Reply: The global MAT should be above 0°C. We incorporated the point in lines 180-181: “The high elevation of the permafrost suggests a warm climate with a global MAT > 0°C during the Early and Middle Noachian epochs.”

Line 159: The authors mention “sand cover increases the amount of Fe” but do not have values to support that. I think this is right but perhaps the manuscript could benefit from including values. One suggestion would be to cite the composition of Fe in sand vs. dust. Maybe use some examples from the rover and global dust values?

Reply: Thank you for your comment regarding the statement about sand cover increasing the amount of Fe. We appreciate your suggestion to provide values to support this claim. In response, we have added the average Fe abundance of dust from rover data in lines 199-202: “They are minimum values of Fe depletion because the covering of sand and dust, which has higher average Fe abundance (~14.9 wt.%)⁵⁰ than Noachian surface average (~14.3 wt. %) on these terrains, would cause the actual depletion values to be underestimated.”

Line 172: The authors state “sand cover does not affect the observed trend of decreasing Fe leaching intensity over time”, however, could this trend also be because older sands have more time to accumulate dust?

Reply: Thank you for your comment regarding the potential impact of sand cover on the observed trend of decreasing Fe leaching intensity over time. We understand your concern and agree that older sands may have more time to accumulate dust, which could affect the observed trend. To address this issue, we have modified the sentence in Lines 202-206: “Because older terrains might host more of these materials due to longer time of sand/dust accumulation and larger and more craters, the amount of surface Fe depletion of the older Noachian epochs is possibly more underestimated. Therefore, excluding the sand/dust cover would strengthen the observed trend of decreasing Fe leaching intensity over time during the Noachian period (Fig. 3a).” This modification clarifies our argument and emphasizes the robustness of the observed trend, even when considering the potential influence of sand and dust accumulation.

Figure 1: the sulfates are not really discussed all that much in text. I suggest adding a couples lines in the text to warrant it’s importance or delete the sulfates in this map

because my eyes are drawn there.

Reply: Sulfate might be products of seasonal freeze-thaw cycles according to our recent publication¹⁰. To explain more about it, a new paragraph was added in lines 284-301: “The seasonal freeze-thaw region has transitioned from high-elevation areas to high-latitude regions as the climate mode has shifted with the gradual loss of H and H₂ in the atmosphere. With further cooling of Mars after NHT, seasonal freeze-thaw cycles could have migrated to low-latitude region during the Hesperian period. Owing to drying of Noachian Mars by sequestration of water in the crust¹¹, Hesperian Mars was arid with much less top down-leaching. Cryosuction during freezing can induce accumulation of water and ions from groundwater in the near-surface environments¹². During the thawing stage, strong evaporation under oxidizing conditions during the Hesperian period, can precipitate salts and ferric oxides⁶⁷. The enrichment of sulfates at Meridiani, the landing site of the Opportunity rover, can be well explained by seasonal freeze-thaw cycles¹⁰ according to the geochemical trends and textural observations. The evidence includes upwards enrichment of S but depletion of Cl, the upper bleaching and brecciated zone and lower darker and less brecciated zone with a sudden horizontal boundary in a meter-scale section, and the platy structure with abundant coarse planar voids but little fine voids. During the Hesperian period, the intersecting regions with high surface temperature (low latitude region) and a shallow water table promoted seasonal freeze-thaw with upward migration of groundwater via cryosuction, thus providing favorable conditions for the formation of sulfates¹⁰. This mechanism may explain why sulfates (blue squares in Fig. 1b) tend to be enriched in low latitude regions.”

Figure 3b: migration to poles vs. polars? Or polar regions vs. polars?

Reply: We modified “polars” to “polar regions” here.

Check all figure numbers and references to figures throughout.

Reply: We have carefully reviewed all figure numbers and references to figures throughout the manuscript to ensure their accuracy and consistency.

References:

1. Niles, P. B., Michalski, J., Ming, D. W. & Golden, D. C. Elevated olivine weathering rates and sulfate formation at cryogenic temperatures on Mars. *Nature Communications* vol. 8 Preprint at <https://doi.org/10.1038/s41467-017-01227-7> (2017).
2. Niles, P. B., Michalski, J., Ming, D. W. & Golden, D. C. Elevated olivine weathering rates and sulfate formation at cryogenic temperatures on Mars.

- Nature Communications* vol. 8 Preprint at <https://doi.org/10.1038/s41467-017-01227-7> (2017).
3. Zandanel, A. *et al.* Geologically rapid aqueous mineral alteration at subfreezing temperatures in icy worlds. *Nat Astron* (2022) doi:10.1038/s41550-022-01613-2.
 4. Grau Galofre, A., Jellinek, A. M. & Osinski, G. R. Valley formation on early Mars by subglacial and fluvial erosion. *Nat Geosci* **13**, (2020).
 5. Boatwright, B. D. & Head, J. W. A Noachian proglacial paleolake on Mars: Fluvial activity and lake formation within a closed-source drainage basin crater and implications for early Mars climate. *Planetary Science Journal* **2**, (2021).
 6. Kong, P., Ebihara, M. & Palme, H. Siderophile elements in Martian meteorites and implications for core formation in Mars. *Geochim Cosmochim Acta* **63**, (1999).
 7. Gleason, J. D., Kring, D. A., Hill, D. H. & Boynton, W. V. Petrography and bulk chemistry of Martian orthopyroxenite ALH84001: Implications for the origin of secondary carbonates. *Geochim Cosmochim Acta* **61**, (1997).
 8. Fan, B., Jansen, M. F., Mischna, M. A. & Kite, E. S. Why are mountaintops cold? The transition of surface lapse rate on dry planets. *Geophys Res Lett* **50**, e2023GL106683 (2023).
 9. Kite, E. S. *et al.* Changing spatial distribution of water flow charts major change in Mars's greenhouse effect. *Sci Adv* **8**, eabo5894 (2022).
 10. Jiacheng Liu, Joseph R. Michalski, Wenyuan Gao, Christian Schröder & Yi-Liang Li. Freeze-thaw cycles drove chemical weathering and enriched sulfates in the Burns formation at Meridiani, Mars. *Sci Adv* (2024).
 11. Scheller, E. L., Ehlmann, B. L., Hu, R., Adams, D. J. & Yung, Y. L. Long-term drying of Mars by sequestration of ocean-scale volumes of water in the crust. *Science (1979)* (2021).
 12. Xie, H.-Y. *et al.* Interaction of soil water and groundwater during the freezing–thawing cycle: field observations and numerical modeling. *Hydrol Earth Syst Sci* **25**, 4243–4257 (2021).
 13. Kite, E. S. & Conway, S. Geological evidence for multiple climate transitions on Early Mars. *Nat Geosci* 1–10 (2024).
 14. Kite, E. S. Geologic constraints on early Mars climate. *Space Sci Rev* **215**, 1–47 (2019).
 15. Urata, R. A. & Toon, O. B. Simulations of the martian hydrologic cycle with a general circulation model: Implications for the ancient martian climate. *Icarus* **226**, (2013).
 16. Head, J. W., Wordsworth, R. D. & Fastook, J. L. When Did Mars Become

- Bipolar?: An Analysis of the Key Factors in the Late Noachian-Amazonian Climate Transition from an Altitude-Dominant Temperature Distribution (ADD) to a Latitude-Dominant Distribution (LDD). in *Seventh International Workshop on the Mars Atmosphere: Modelling and Observations* 4305 (2022).
17. Tutolo, B. M. & Tosca, N. J. Observational constraints on the process and products of Martian serpentinization. *Sci Adv* **9**, eadd8472 (2023).
 18. Tutolo, B. M. & Tosca, N. J. Observational constraints on the process and products of Martian serpentinization. *Sci Adv* **9**, eadd8472 (2023).
 19. Wordsworth, R. *et al.* Global modelling of the early martian climate under a denser CO₂ atmosphere: Water cycle and ice evolution. *Icarus* **222**, 1–19 (2013).
 20. Turbet, M. & Forget, F. 3-D Global modelling of the early martian climate under a dense CO₂+ H₂ atmosphere and for a wide range of surface water inventories. *arXiv preprint arXiv:2103.10301* (2021).
 21. Wordsworth, R. D. The Climate of Early Mars. *Annu Rev Earth Planet Sci* **44**, 381–408 (2016).
 22. Boynton, W. V *et al.* Concentration of H, Si, Cl, K, Fe, and Th in the low-and mid-latitude regions of Mars. *J Geophys Res Planets* **112**, (2007).
 23. Hahn, B. C. *et al.* Mars Odyssey Gamma Ray Spectrometer elemental abundances and apparent relative surface age: Implications for Martian crustal evolution. *Journal of Geophysical Research E: Planets* **112**, 1–13 (2007).
 24. Rye, R. & Holland, H. D. Paleosols and the evolution of atmospheric oxygen: A critical review. *American Journal of Science* vol. 298 621–672 Preprint at <https://doi.org/10.2475/ajs.298.8.621> (1998).
 25. Sakhabutdinov, R. Z., Tronov, V. P. & Shavaleev, I. I. Influence of temperature on the rate of oxidation of the Fe (II)-EDTA complex by atmospheric oxygen. *J. Appl. Chem. USSR (Engl. Transl.); (United States)* **58**, (1986).
 26. Fairén, A. G. *et al.* Cold glacial oceans would have inhibited phyllosilicate sedimentation on early Mars. *Nat Geosci* **4**, 667–670 (2011).
 27. Carter, J., Loizeau, D., Mangold, N., Poulet, F. & Bibring, J.-P. Widespread surface weathering on early Mars: A case for a warmer and wetter climate. *Icarus* **248**, 373–382 (2015).
 28. Ehlmann, B. L. & Edwards, C. S. Mineralogy of the Martian Surface. *Annu Rev Earth Planet Sci* **42**, 291–315 (2014).
 29. Tanaka, K. L. *et al.* *Geologic Map of Mars. Scientific Investigations Map* (2014) doi:10.3133/sim3292.
 30. Hahn, B. C. *et al.* Mars Odyssey Gamma Ray Spectrometer elemental abundances and apparent relative surface age: Implications for Martian crustal

- evolution. *Journal of Geophysical Research E: Planets* **112**, 1–13 (2007).
31. Wordsworth, R. *et al.* Global modelling of the early martian climate under a denser CO₂ atmosphere: Water cycle and ice evolution. *Icarus* **222**, 1–19 (2013).
 32. Wordsworth, R. *et al.* A coupled model of episodic warming, oxidation and geochemical transitions on early Mars. *Nat Geosci* **14**, 127–132 (2021).
 33. Rapin, W. *et al.* Sustained wet–dry cycling on early Mars. *Nature* **620**, (2023).
 34. Vaniman, D. T. *et al.* Mineralogy of a mudstone at Yellowknife Bay, Gale crater, Mars. *Science (1979)* **343**, (2014).
 35. Bristow, T. F. *et al.* Brine-driven destruction of clay minerals in Gale crater, Mars. *Science (1979)* **373**, 198–204 (2021).
 36. Bishop, J. L. *et al.* Surface clay formation during short-term warmer and wetter conditions on a largely cold ancient Mars. *Nat Astron* **2**, (2018).

Reviewer #2 (Remarks to the Author):

Please check the attached comments.

Reviewer #2 Attachment on the following page

Summary:

The manuscript has been much improved in data analysis, data visualization, providing the background to a broad audience, and citations. I don't think there is there is more work to be done with the major results, but I still have concerns about some of the logic.

1. In the revised manuscript, the authors clarified that Fe depletion is defined by comparing to the adjacent unit for Early Noachian and Middle Noachian, or global surface average for Late Noachian. It remains unclear to me why it is defined differently for these different time units. Why do the authors not always compare to the global average?
2. In the supplementary information, the authors decomposed data into three latitude bands, and they found the south high latitude data reduces the correlation for Early Noachian and Middle Noachian (Fig. S3a & S5). They attributed the lack of correlation to the permafrost hypothesis and obliquity variations (Line 174-176, Line 232-235).

While the relation with permafrost scenario makes sense, the relation with obliquity does not. Transition from low to high obliquities only reduces the equator-to-pole temperature gradient, which works equally to both northern and southern hemisphere. Moreover, the southern data does not show a negative correlation even if you exclude the elevations above +3 km and below -3 km.

Minor comments:

Line 100: Typo "> 20 deg S"

Line 172-174: This sentence is a bit hard to read. Suggest modifying as "The minimal Fe leaching in permafrost can explain the weak correlation between iron depletion and elevation in southern high latitudes."

Manuscript Number: NCOMMS-23-54917B

Title: Atmospheric Oxidation Drove Climate Change on Noachian Mars

In the document, the **green text** represents our responses, while the **red text** indicates the corresponding changes made in the revised manuscript.

Reviewer Comments

Summary: The manuscript has been much improved in data analysis, data visualization, providing the background to a broad audience, and citations. I don't think there is there is more work to be done with the major results, but I still have concerns about some of the logic.

Reply: We greatly appreciate your positive feedback and the concerns you have raised. In response, we made revisions to address and clarify the logic you commented. We believe that this version presents a clearer and more coherent logical framework.

1. In the revised manuscript, the authors clarified that Fe depletion is defined by comparing to the adjacent unit for Early Noachian and Middle Noachian, or global surface average for Late Noachian. It remains unclear to me why it is defined differently for these different time units. Why do the authors not always compare to the global average?

Reply: Thank you for the concern. The difference between Fe abundance of the Early Noachian terrains and the global average (the potential original Fe abundance of the Noachian crust, 14.3 wt. %) represents the abundance of Fe leaching throughout the entire Noachian period rather than solely during the Early Noachian epoch.

To determine the abundance of Fe leaching specifically in the Early Noachian epoch, we can only use the Fe abundance of the Middle Noachian terrains to subtract the Fe abundance of Early Noachian terrains. We use the same method to calculate the abundance of Fe leaching in the Middle Noachian epoch.

For determining the abundance of Fe leaching in the Late Noachian epoch, we cannot simply subtract the Fe abundance of Early Noachian terrains from that of the Noachian-Hesperian terrains, as there is a noticeable Fe enrichment in the Noachian-Hesperian terrains compared to the Late Noachian terrains and the Early Hesperian terrains. Using this approach would significantly overestimate the Fe leaching intensity. Therefore, to estimate the abundance of Fe leaching in the Late Noachian epoch, we should subtract the Fe abundance of Late Noachian terrains from the potential original Fe abundance of the Noachian crust directly.

To clarify it, we made corresponding revisions in lines 195-205 "the difference (~0.10 wt.%) between this value and the Fe abundance of the Late Noachian terrains can represent the Fe leaching intensity during the Late Noachian epoch (Fig. 3b). However, for determining the abundance of Fe leaching specifically in the Early Noachian epoch, we cannot simply use the difference between Fe abundance of the Early Noachian terrains and the potential original Fe abundance of the Noachian crust, as this would represent the abundance of Fe leaching throughout the entire Noachian period rather than solely during the Early Noachian epoch. To determine the abundance of Fe leaching specifically in the Early Noachian epoch, we must subtract the Fe abundance of the Middle Noachian terrains from that of Early Noachian terrains. We employ the same method to calculate the abundance of Fe leaching in the Middle

Noachian epoch. The Fe leaching intensity in the Early and Middle Noachian epochs are approximately 0.58 wt.% and 0.15 wt.%, respectively (Fig. 3b).” and lines 547-549: “The abundance of Fe leaching in each epoch is calculated based on the value of Fe depletion relative to adjacent unit (for Early Noachian and Middle Noachian epoch) or the potential original Fe abundance of the Noachian crust (for Late Noachian epoch).”

2. In the supplementary information, the authors decomposed data into three latitude bands, and they found the south high latitude data reduces the correlation for Early Noachian and Middle Noachian (Fig. S3a & S5). They attributed the lack of correlation to the permafrost hypothesis and obliquity variations (Line 174-176, Line 232-235). While the relation with permafrost scenario makes sense, the relation with obliquity does not. Transition from low to high obliquities only reduces the equator-to-pole temperature gradient, which works equally to both northern and southern hemisphere. Moreover, the southern data does not show a negative correlation even if you exclude the elevations above +3 km and below -3 km.

Reply: Thanks for the comment. We agree that transition from low to high obliquities only reduces the equator-to-pole temperature gradient. The lack of correlation between elevation and Fe abundance in the southern high latitude region is not related to the obliquity. As such, we removed the statements regarding obliquities in the two sentences mentioned by you. We believe that this revision makes the logic clearer in the current version.

Minor comments:

Line 100: Typo “> 20 deg S”

Reply: Done.

Line 172-174: This sentence is a bit hard to read. Suggest modifying as “The minimal Fe leaching in permafrost can explain the weak correlation between iron depletion and elevation in southern high latitudes.”

Reply: Thank you for pointing it out. We have made the modifications according to your suggestion in lines 170-172 “The minimal Fe leaching in permafrost can explain the lack of correlation between iron depletion and elevation in southern high latitudes (Supplementary Fig. 4).”